# Dataflow-Guided Neuro-Symbolic Language Models for Type Inference

**Gen Li**[1] **Yao Wan**[1] **Hongyu Zhang**[2] **Zhou Zhao**[3] **Wenbin Jiang**[1] **Xuanhua Shi**[1] **Hai Jin**[1] **Zheng Wang**[4]

## Abstract

*Language Models* (LMs) are increasingly used for type inference, aiding in error detection and software development. Some real-world deployments of LMs require the model to run on local machines to safeguard the intellectual property of the source code. This setting often limits the size of the LMs that can be used. We present NESTER, the first neuro-symbolic approach that enhances LMs for type inference by integrating symbolic learning without increasing model size. NESTER breaks type inference into sub-tasks based on the data and control flow of the input code, encoding them as a modular high-level program. This program executes multi-step actions, such as evaluating expressions and analyzing conditional branches of the target code, combining static typing with LMs to infer potential types. Evaluated on the Many-Types4Py dataset in Python, NESTER outperforms two state-of-the-art type inference methods (Hi-Typer and TypeGen), achieving 70.7% Top-1 Exact Match, which is 18.3% and 3.6% higher than HiTyper and TypeGen, respectively. For complex type annotations like "typing.Optional" and "typing.Union", NESTER achieves 51.0% and 16.7%, surpassing TypeGen by 28.3% and 5.8%.

## 1. Introduction

Dynamically typed languages like Python and JavaScript are popular for rapid prototyping and have the flexibility of not requiring explicit type declarations (Mir et al., 2022; Srinath, 2017). However, this flexibility can compromise reliability, often leading to runtime type errors that are diffi-

---

[1]National Engineering Research Center for Big Data Technology and Systems, Services Computing Technology and System Lab, Cluster and Grid Computing Lab, School of Computer Science and Technology, Huazhong University of Science and Technology, Wuhan, China [2]Chongqing University [3]Zhejiang University [4]University of Leeds. Correspondence to: Yao Wan <wanyao@hust.edu.cn>.

*Proceedings of the 42$^{nd}$ International Conference on Machine Learning*, Vancouver, Canada. PMLR 267, 2025. Copyright 2025 by the author(s).

cult to debug (Oh & Oh, 2022). Therefore, automated type inference, which infers the type of a variable automatically, is attractive. Recently, there has been growing interest in using *Language Models* (LMs) for type inference to reduce the manual effort in developing type inference systems.

*Large Language Models* (LLMs), such as GPT-4 (Koubaa, 2023) and OpenAI-o1 (Temsah et al., 2024), have shown impressive results in type inference. However, these models typically rely on cloud-based infrastructure with substantial computational resources, raising privacy concerns when user data is sent to untrusted providers (Ray, 2023). This is especially problematic for organizations that treat their source code as valuable intellectual property. To mitigate risks, many prefer deploying smaller models locally on developer machines or private clusters due to hardware resource constraints.

Running moderately sized LMs, like Code Llama with 7B parameters (Lu et al., 2023), locally can address privacy and confidentiality concerns. It also delivers faster response times compared to cloud solutions employing fully homomorphic encryption (Chen et al., 2022) or multi-party communication techniques (Yao, 1986; Rathee et al., 2024). However, downsizing LMs from hundreds of billions to tens of billions of parameters significantly reduces their ability to model and reason about complex program structures. As a result, there is a need to improve the capability of LMs in reasoning code for type inference without relying on cloud-based solutions.

**A Motivation Example** Figure 1 illustrates a motivating example for inferring the return type of the Python function learning_rate. As shown in Figure 1(a), mypy[1], a rule-based tool, fails to infer the type correctly due to its lack of knowledge of custom functions or third-party libraries (*e.g.*, _normalize_lr in line 7). Similarly, a pretrained CodeLlama 7B (Lu et al., 2023) predicts the type as bool—float—none, while the correct type is float—none, as seen in Figure 1(b). This error stems from the LM relying on the textual structure of the code rather than its underlying semantics, such as control and data flow. Additionally, LMs can be misled by irrelevant nearby information (*e.g.*, return False in line 3). Specifically, since lr never exceeds 1.0 and line 1 is unreachable, the return type cannot

---

[1]https://mypy-lang.org/

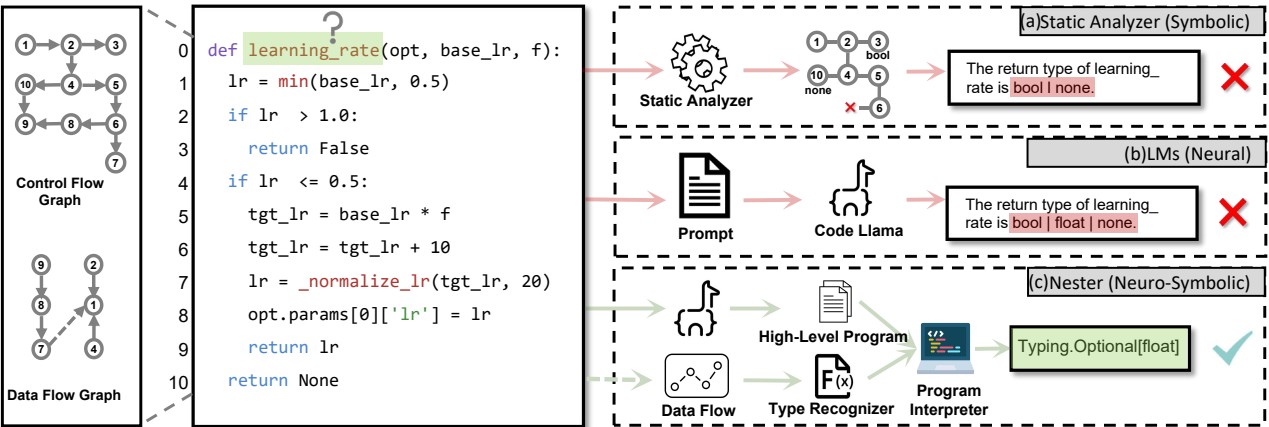

Figure 1. Example showing the 7B CodeLlama LM and static analyzer (mypy) fail to infer the return type of learning_rate.

be boolean. This issue can be addressed by incorporating control flow analysis via static techniques.

**Our Solution: A *Neuro-Symbolic* Approach**   Our work aims to enhance LMs' ability to reason about program control and data flow, essential for accurate type inference, without relying on larger models. We propose a novel neuro-symbolic framework, NESTER, which decomposes type inference into simpler symbolic steps encoded as a Python-like *high-level program*. This high-level program is then processed by a program interpreter by leveraging static type inference and LMs. As shown in Figure 1(c), we first break down type inference into a number of sub-tasks according to control flow, where each sub-task corresponds to a potential type of the execution path. The analysis of the sub-task is realized through a number of API calls like if_analysis (inputs, conditions, ...). Then, a program interpreter processes each line of the high-level program to combine static type inference and LMs to infer the type at each step. Our neuro-symbolic approach also enhances interpretability by embedding reasoning rules into LMs, improving decision-making transparency.

**Key Results and Contributions**   We evaluate NESTER by applying it to the widely-used ManyTypes4Py dataset (Mir et al., 2021). Experimental results show that NESTER outperforms the advanced Type4Py baseline by 3.8% for argument types and 3.9% for return types in Top-1 Exact Match accuracy, using an LM with just 7B parameters. Additionally, NESTER surpasses the state-of-the-art TypeGen baseline by 3.6% across all categories. This demonstrates NESTER's efficiency, achieving strong performance using a small LM by today's standards. Case studies on real-world code snippets further highlight NESTER 's ability to accurately infer types with natural-language explanations.

The key contributions of this paper are as follows:

- We propose NESTER, the first neuro-symbolic framework for type inference using LMs, which enhances reasoning

and explainability by incorporating symbolic rules.
- We introduce data flow analysis of the target program to guide LMs in inferring types more precisely.
- Extensive experiments on ManyTypes4Py demonstrate the effectiveness of NESTER, and a VSCode plugin is developed to showcase its enhanced interpretability.

**Online Materials**   The source code and data associated with this work are available at: https://github.com/CGCL-codes/naturalcc/tree/main/examples/nester.

## 2. Preliminaries

### 2.1. Type Inference

This work focuses on the type inference task, which predicts an identifier's type based on its context in a code snippet. Given a code snippet $c$, let $\{x_v \mid v \in 1 \ldots V\}$ represent the $V$ distinct identifiers needing type assignment. Let $p(\tau_i \mid ctx)$ denote the type distribution for identifier $x_v$, determined by its context $ctx$. Our goal is to develop a model that, given an unannotated or partially annotated code snippet $c$, outputs a type distribution for each missing annotation. Recent studies have used LLMs for type inference (Siddiqui & Kellogg, 2024; Peng et al., 2023). These approaches frame the task as a generative one through natural language prompts, which describe the code and identify relevant identifiers for type prediction. TypeGen (Peng et al., 2023) is the state-of-the-art in LM-based type inference, using a specific prompt based on *In-Context Learning* (ICL). The prompt structure is [Code, ICL EXAMPLES, TYPEHINTS], where ICL EXAMPLES provide demonstrations, and TYPEHINTS are derived from static analysis. TypeGen also uses CoT reasoning to improve accuracy, with the prompt structured as follows: "*First, the variable* error_message *is assigned from a string. Therefore, the type of the variable* error_message *is "str"."*

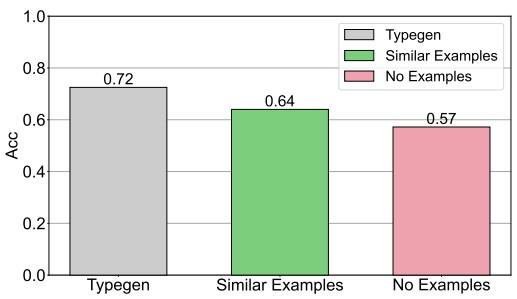

*Figure 2.* The reasoning ability study on TypeGen.

## 2.2. Neuro-Symbolic Reasoning

Neuro-symbolic reasoning enhances interpretability by breaking tasks into subtasks (Gupta & Kembhavi, 2023; Cunnington et al., 2022; Nye et al., 2021; Zhang et al., 2021). It divides the solution function $f$ into two phases:

$$f = h \circ g.$$

Here, $g$ is a perception network that processes raw data, imposing human-like constraints to define conditions for the problem, while $h$ is a reasoning model that analyzes the data to infer the solution.

Consider *Visual Question Answering* (VQA) as an example, where the goal is to answer a question based on an image (Gupta & Kembhavi, 2023). The image is processed into a structured format by the visual perception network $g$, serving as static information for the question. For example, given the query "*How many cubes behind the cylinder are large?*", the context-aware model $h$ parses the question into a high-level program. Key terms like *cube*, *cylinder*, and *large* are extracted as parameters. The program then uses APIs such as *filter_shape* and *count* to generate a solution. Finally, the program is executed using the database from the visual perception network.

## 2.3. Do LMs Demonstrate Reasoning Ability in Type Inference?

Before introducing NESTER, we first conduct a preliminary study to explore whether LMs demonstrate reasoning ability in type inference. While LMs have shown promising results in type inference, we attribute their successful type predictions primarily to the examples provided in ICL demonstrations, rather than to the reasoning over symbolic type rules. To verify our suspicion, we select TypeGen, a state-of-the-art LM-based type inference approach, as our target for analysis, over the widely utilized ManyTypes4Py dataset (Mir et al., 2021). In implementation, we select Code Llama 7B as the LM backbone, and design two settings: (1) *No Examples:* We remove all demonstration examples, keeping only the query code and the question asking for the type of code. (2) *Similar Examples Replacement:* We replace the

demonstration examples, whose type answers are identical to those in the test set, with examples from a pool of candidate inference examples that have differing type answers from the test set. These new demonstration examples, like those selected by TypeGen, exhibit high code similarity to the training set.

Figure 2 shows the performance of TypeGen in type inference over the aforementioned two settings. From this figure, we can observe that incorporating examples with identical result types into TypeGen surely enhanced accuracy, reaching 72.5% compared to 57.2% without any examples, marking an increase of 15.3%. However, after replacing with similar examples, the accuracy of examples provided to LMs decreased to 64%. This marks a reduction of 8.5% from the original TypeGen scenario, effectively halving the initial improvement. The experimental results indicate that LMs do not truly understand the problem; they merely generate text and append the type result at the end, resulting in misleadingly high accuracy. However, when replaced with similar examples, LMs become confused by out-of-distribution test sets, which hampers their ability to perform generalization reasoning and ultimately leads to a halving of accuracy. Consequently, while TypeGen's inference examples misleadingly improve accuracy, they lack reasonableness in the LM type inference process.

## 3. NESTER: Our Approach

Figure 3 provides an overview of our proposed system, NESTER, which comprises two main components: (a) *high-level program generation* and (b) *program interpreter*. The high-level program generation component involves crafting a prompt (*e.g.*, *"Convert the code into a high-level program"*) along with several demonstration examples to guide the LMs in generating high-level programs composed of predefined APIs. In the program interpreter component, we sequentially implement each API derived from the high-level programs. Each API utilizes a tandem of the type recognizer and LMs to perform type inference by traversing the dataflow starting from the API's parameter.

### 3.1. High-Level Program Generation

NESTER first decomposes the code snippet $c$ into a high-level program, which represents reasoning substeps in the structure of the code. Each line of the high-level program is an API call to a function supported by NESTER, which is then processed by our program interpreter. The main function of the high-level program is guiding the program interpreter to focus on specific variables and execution paths, while the API performs detailed analysis on a variable, conditions, or a string. It is noteworthy that we preserve the original program's control flow information based on the high-level program. On the one hand, the high-level pro-

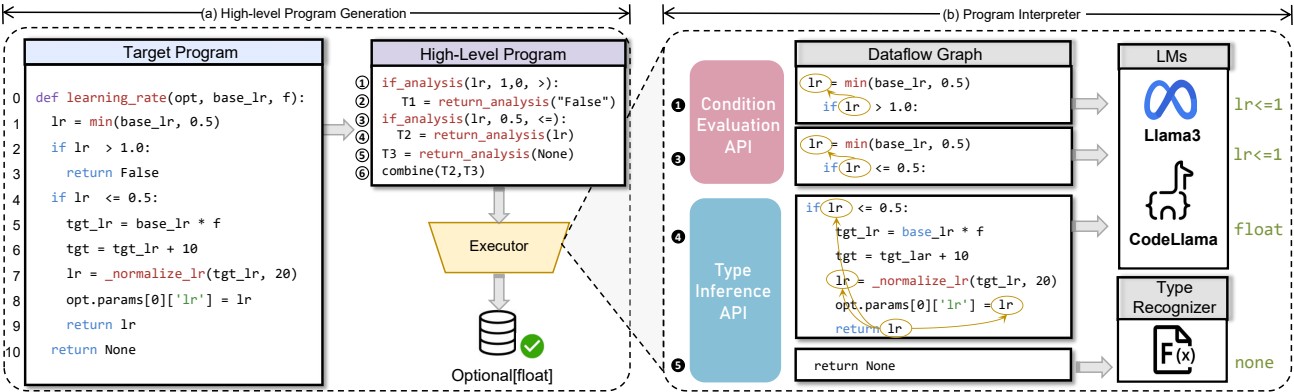

*Figure 3.* Neuro-symbolic type inference pipeline. blue highlights the target program, purple highlights the high-level program generated by LMs, and green highlights the output by our program interpreter. ① to ⑥ are the line numbers of the high-level program. ❶, ❸ are executed by the condition evaluation API. ❹, ❺ are executed by the type inference API.

gram can utilize the type information implicitly contained in the control flow; on the other hand, it contributes to the final type inference results combination. A running example of the high-level program is given in Figure 3. Moreover, we also provide a concrete definition of the high-level program, as detailed in Appendix B.1.

**A Running Example**   Figure 3(a) shows an example high-level program generated by NESTER for the target program in Figure 1. Here, we generate the high-level program by prompting an LM (*e.g.*, Code Llama 7B), although this can also be done through a traditional parser. We use LMs because a general-purpose LM can adapt to multiple programming languages, reducing the manual efforts in building a language-specific parser. We use a prompt consisting of code and high-level program pairs to enable LMs to understand the translation task. For details on the specific prompt, please refer to Appendix B.2.

NESTER generates the high-level program by translating each conditional branch of the input code into an analysis unit in the high-level program, realized through API calls. We do so by providing examples in the LM prompt to ask it to generate API calls for a given code input. For the example given in Figure 3(a), we start by feeding `if lr > 1.0` at line 2 to an LM to generate an API call, `if_analysis`. The source `if` expression is broken down into identifiers, operators, and literal values: `lr`, `1.0`, and `>`. Similarly, `if lr <= 0.5` is also parsed by the LM to `lr`, `0.5`, and `<=`. For the code in lines 3, 9, and 10, the LM parses this code as `return False` to `T2 = return_analysis(False)`, `return lr` can transform to `T2 = return_analysis(lr)` and `return None` to `return_analysis(None)`, respectively. The API calls in the high-level program guide NESTER for type inference, where each analysis unit essen-

tially corresponds to a type inference sub-task.

**Supported APIs**   NESTER offers two types of APIs to be used in the high-level programs: (i) *Condition evaluation APIs* to evaluate if a condition in the input program can be satisfied to determine the reachable execution paths; (ii) *Type inference entry-point APIs* to retain identifiers (*e.g.*, `lr` in Figure 1) that guard the evaluation outcomes of a condition. NESTER is extensible - new APIs can be added by creating and registering a class. The current implementation of NESTER provides four APIs, detailed in Appendix B.3.

### 3.2. Program Interpreter

The high-level program is interpreted and executed by a NESTER program interpreter, which takes two inputs: (1) the target program for type inferencing and (2) a high-level program generated from the target program. The program interpreter first performs data flow analysis on the target program to construct a data flow graph. Then, we traverse the dataflow graph, beginning at the entry point of each API call of the high-level program.

**Dataflow Graph**   Figure 4 illustrates an example of constructing the dataflow graph for the target program. Specifically, for the identifier `lr`, we first locate the initial definition in the code snippet, namely `lr = min(base_lr, 0.5)` on line 1. Consequently, we add the line as a node to **Def**($lr$), denoted as **Def**($lr$) = {Line 1}. After defining **Def**($lr$), We proceed to identify the lines of code that directly use each member belonging to **Def**($lr$). These are the **User**($lr$) nodes, explicitly denoted as {Lines 2, 4, 7, 8 and 9}. After identifying all the **User**($lr$) nodes, we apply the same method to find the **Usee**($lr$) nodes, which capture the lines of code that utilize the users from both **User**($lr$) and **Def**($lr$), denoted as {Lines 0, 5}. We use the *Coarse-Grained Dataflow Graph* (CGDG) as the implementation of this data flow graph. For details of the

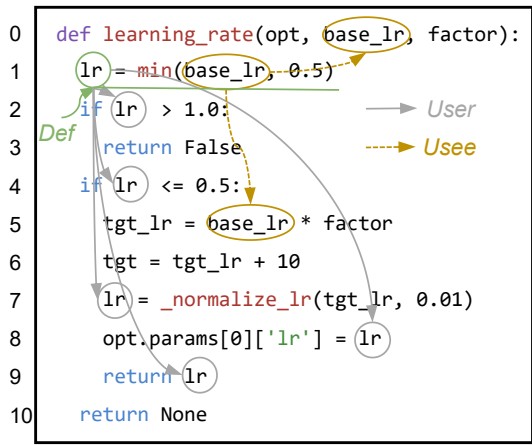

```
0    def learning_rate(opt, base_lr, factor):
1        lr = min(base_lr, 0.5)
2        if lr > 1.0:
3            return False
4        if lr <= 0.5:
5            tgt_lr = base_lr * factor
6            tgt = tgt_lr + 10
7            lr = _normalize_lr(tgt_lr, 0.01)
8            opt.params[0]['lr'] = lr
9            return lr
10       return None
```

| Identifier | Def | User | Usee |
|---|---|---|---|
| opt | line 0 | line 1,5 | - |
| base_lr | line 0 | line 1,5 | - |
| factor | line 0 | line 5 | - |
| lr | line 1 | line 2,4,7,8,9 | line 0,5 |
| tgt_lr | line 5 | line 6,7 | line0 |

T2 = return_analysis(lr✗)    :float✓

*Figure 4.* The construction of a dataflow graph for a code snippet.

CGDG construction, please refer to Appendix C.2.

**Type Recognizer** NESTER uses the type recognizer to identify the types of a specific node (which include multiple code lines) in dataflow graph, including $\mathbf{Def}(x_v)$, $\mathbf{User}(x_v)$, and $\mathbf{Usee}(x_v)$. If a line of code contains type information, the type recognizer uses regular expressions to recognize and return the corresponding type. These types include basic types such as "int", "float", "bool", "str", and "bytes", as well as more complex generic types such as "tuple", "list", and "dict". Appendix C.1 presents the denotational semantics of the regular expressions, which is used by the type recognizer in NESTER.

**LMs System** NESTER employs two LMs for program interpretation: the Condition Eval LM and the Type Infer LM. The Condition Eval LM assesses the possible value ranges for specified identifiers, while the Type Infer LM identifies the type of given code snippets, as a supplement for the type recognizer. We utilize prompts to elicit responses from the LLM as follows: [$\mathbf{Def}(x_v) \cup \mathbf{User}(x_v) \cup \mathbf{Usee}(x_v)$(Code), QUESTION_CONDITION] and [$\mathbf{Def}(x_v) \cup \mathbf{User}(x_v) \cup \mathbf{Usee}(x_v)$(Code), TYPEHINTS, QUESTION_TYPE], respectively.

Figure 3(b) shows an example of a program interpreter. In terms of condition evaluation, API execution. ❶It looks for identifiers in the API call of the high-level program and uses data flow analysis on the target program to locate relevant code segments

`learning_rate = min(base_lr, 0.5)` and consults an LM to determine the range of values for `lr`. In this case, the LM returns the range `lr <= 0.5`, which is a correct return result. Then we utilize the results to evaluate the condition `lr > 1.0`, which is not satisfied, so we jump over ② to execute ③ in the high-level program. ❸ The code `if lr <= 0.5` is also parsed by the LM into `if_analysis(lr, 0.5, <=)`. Using the result `lr <= 0.5`, we evaluate that $lr \leq 0.5$ holds true. ❹ We utilize a dataflow graph to identify code lines in the target program, which are related to `lr`. These lines are initially sent to a type recognizer. Upon failure, the lines are forwarded to LMs, which then return the type "float". ❺ The type recognizer directly returns the type "none" to our API. ❻ Finally, the API `Combine` integrates the two types, "float" and "none", based on the return values provided by `return_analysis`. Consequently, it derives the final correct type "Optional[float]".

Note: When handling multiple `if_analysis` APIs, the `combine()` module detects and discards unreturned type T before merging the remaining types.

## 4. Experiments

### 4.1. Evaluation Setup

**Dataset** We follow the methodology from previous studies (Allamanis et al., 2020; Peng et al., 2023; Lukasczyk et al., 2023; Zhang et al., 2023) and evaluate our approach using the widely-used ManyTypes4Py dataset (Mir et al., 2021), which contains around 5,500 Python projects with over 870,000 type annotations. The dataset includes over 880,000 functions (return types), 1.5 million arguments (argument types), and 2.1 million variable declarations (local types). To ensure fairness in comparison, we normalize outputs by treating similar terms, such as "int" and "integer" or "bool" and "boolean" as equivalent.

**LM Backbones** We select two LMs as backbones.

▷ **Code Llama 7B** (Lu et al., 2023) is an LLM specifically designed for code-related tasks. This variant, with 7 billion parameters, is pre-trained on a diverse set of code and natural language data, enabling it to effectively understand and generate code across multiple programming languages. The pre-training process includes extensive datasets from open-source repositories, which helps Code Llama 7B develop a robust understanding of syntax, structure, and various programming paradigms.

▷ **Llama 3 8B** (Das et al., 2025b) is a versatile foundation model designed for multilinguality, coding, reasoning, and tool usage. This variant, with 8 billion parameters, leverages a dense Transformer architecture, enabling it to handle com-

plex tasks across different languages and domains. Llama 3 8B is pre-trained on diverse data sources, including text, code, and structured information, which enhances its capability to perform well in various applications, from language understanding to code generation.

**Baselines**   We compare NESTER with eight baselines.

▷ **Hityper** (Peng et al., 2022) combines static analysis and well-designed static rules for type inference.

▷ **TypeBERT** (Jesse et al., 2021) treats type inference as a fill-in-the-blank task, pre-training a BERT-style model on a large-scale code corpus for this purpose.

▷ **TypeWriter** (Pradel et al., 2020) applies a classification method for type inference. It segments code into identifiers, tokens, comments, and types, then uses a sequence-to-sequence RNN model (Hochreiter & Schmidhuber, 1997) for classification.

▷ **UniXcoder** (Guo et al., 2022) is a pre-trained model for both code understanding and generation, applied to type inference tasks.

▷ **Type4Py** (Mir et al., 2021) employs clustering for type inference by mapping types into continuous space, allowing rare types to be predicted by analyzing cluster distances.

▷ **InCoder** (Fried et al., 2022) is a code generation LM that treats type inference as a cloze-style task, completing type annotations within code segments.

▷ **CodeT5** (Wang et al., 2021) is an LM for type inference, using the T5 architecture and fine-tuned on code datasets to generate type annotations with improved accuracy.

▷ **TypeGen** (Peng et al., 2023) uses a generative approach for type inference via the chain-of-thought method. It structures the prompt into code snippets, type hints, questions, and answers to guide LMs in generating type inference responses.

**Evaluation Metrics**   Following previous studies (Allamanis et al., 2020; Mir et al., 2021; Peng et al., 2023), we adopt two evaluation metrics: Exact Match and Match to Parametric. These metrics assess the proportion of outcomes that: 1) Exact Match: fully correspond to human annotations, for example, "Dict{int: int}" matches exactly with "Dict{int: int}" and "Dict{str: str}" with "Dict{str: str}"; and 2) Match to Parametric: meet the criteria of an exact match, excluding type parameters, thus treating types like "Dict{int: int}" and "Dict{str: str}" as equivalent to a generic "dict" under this metric. The experiment utilizes Top-1, Top-3, and Top-5 accuracy metrics to assess the performance comprehensively.

**Implementation Details**   The model is configured with a 4,096-token context window and a maximum batch size of 8, which is adequate for most dynamic type inference tasks. To improve accuracy while keeping computational demands manageable, we set the temperature to 0.2. This configuration minimizes output randomness, enabling consistent and reliable type inference across diverse coding scenarios. All experimental tasks are carried out on one Linux platform, running an Ubuntu 22.04 system, powered by an Intel Xeon 112-core CPU running at 2.40GHz, coupled with an RTX 4090 GPU, and equipped with 1 TB of RAM.

### 4.2. Overall Performance

To evaluate NESTER 's performance, we compare it against several baselines: a symbolic method (HiTyper), four deep learning models (TypeBERT, TypeWriter, UniXcoder, Type4Py), and eight LM-based methods (InCoder [1.3B, 6.7B], CodeT5 [base, large], Code Llama, Llama3, TypeGen). Table 1 presents the results based on Exact Match and Match to Parametric metrics. NESTER is competitive with the top-performing deep-learning method, Type4Py, across nearly all Match to Parametric categories (Top-1 to Top-5). Specifically, NESTER scores 81.5% for Top-1, surpassing Type4Py's 80.2%. In Exact Match, NESTER exceeds Type4Py by 3.8% and 3.9% for arguments and returns, respectively. In Match to Parametric, NESTER outperforms Type4Py by 7.7% and 5.1%. Interestingly, NESTER lags behind Type4Py in Exact Match for both local and overall evaluations, due to Type4Py's strength in handling short code segments, despite its limited reasoning ability. However, NESTER 's performance is based on 7B-parameter LMs, and we expect further improvements with larger models. We also compare NESTER with several LM-based approaches, including TypeGen, InCoder (1.3B, 6.7B), UniXcoder, CodeT5 (base, large), Code Llama (7B), and Llama3 (8B). NESTER generally outperforms these methods, achieving 70.7% accuracy for Top-1 Exact Match, surpassing TypeGen by 3.6%. It also exceeds TypeGen across all metrics, with a 5.3% improvement in Match to Parametric. This highlights the benefit of combining LM generative capabilities with the reasoning strengths of static rules for type inference.

### 4.3. Effectiveness of Neural Understanding

To explore how NESTER learns control flow via neural understanding, we compare it with TypeGen, focusing on the typing module, which humans use for optional typing. Figure 5 shows the precision of NESTER and TypeGen across different typing categories: "Optional", "Union", "Pattern", "Match", "Defaultdict", "Type", "Deque", "IO". In common categories like "typing.Optional" and "typing.Union", NESTER achieves precision scores of 51.0% and 16.7%, significantly outperforming TypeGen (22.7%

*Table 1.* The performance of NESTER along with the baselines under four types of variables in terms of Top-1,3,5 Exact Match (%) and Match to Parametric (%).

| Metric | Cat. | Approach | Top-1 (%) | | | | Top-3 (%) | | | | Top-5 (%) | | | |
|---|---|---|---|---|---|---|---|---|---|---|---|---|---|---|
| | | | Arg. | Ret. | Var. | All | Arg. | Ret. | Var. | All | Arg. | Ret. | Var. | All |
| **Exact Match (%)** | ST | Hityper | 8.0 | 43.5 | 65.7 | 52.4 | 8.0 | 43.5 | 65.7 | 52.4 | 8.0 | 43.5 | 65.7 | 52.4 |
| | DL | TypeBERT | 28.0 | 38.5 | 51.1 | 45.4 | 34.8 | 52.6 | 55.8 | 51.4 | 36.5 | 57.1 | 58.6 | 54.1 |
| | | TypeWriter | 53.3 | 52.8 | - | - | 61.1 | 60.7 | - | - | 65.8 | 65.3 | - | - |
| | | UniXcoder | 55.0 | 49.2 | 35.9 | 40.9 | 66.9 | 64.6 | 42.1 | 49.0 | 70.6 | **69.8** | 45.2 | 52.4 |
| | | Type4Py | 66.5 | 56.1 | **82.0** | **76.6** | 72.0 | 59.2 | **83.8** | **79.3** | 73.8 | 60.7 | **84.3** | **80.1** |
| | LMs | InCoder-1.3B | 20.9 | 20.5 | 15.1 | 16.7 | 21.3 | 20.8 | 15.5 | 17.1 | 21.3 | 21.0 | 15.6 | 17.2 |
| | | InCoder-6.7B | 24.1 | 42.0 | 18.7 | 21.9 | 24.6 | 42.7 | 19.1 | 22.3 | 24.7 | 43.1 | 19.2 | 22.4 |
| | | CodeT5-base | 51.1 | 57.6 | 21.7 | 30.7 | 59.3 | 64.4 | 28.0 | 37.4 | 62.0 | 66.9 | 30.7 | 40.1 |
| | | CodeT5-large | 56.2 | **60.2** | 44.7 | 48.4 | 61.6 | 64.5 | 50.4 | 53.9 | 63.9 | 66.3 | 53.4 | 56.6 |
| | | Naive-CL | 33.5 | 22.1 | 32.2 | 31.7 | 48.9 | 35.1 | 46.2 | 46.0 | 52.8 | 36.4 | 49.8 | 49.5 |
| | | Naive-L3 | 31.3 | 41.6 | 40.2 | 38.3 | 45.9 | 49.4 | 55.0 | 52.3 | 46.0 | 49.4 | 56.3 | 53.4 |
| | | TypeGen-CL | 61.8 | 58.0 | 69.7 | 67.1 | 71.5 | 61.0 | 76.6 | 74.5 | 75.9 | 65.3 | 77.4 | 76.3 |
| | | TypeGen-L3 | 56.9 | 52.2 | 60.5 | 59.0 | 69.8 | 65.6 | 73.6 | 72.4 | 71.5 | 68.2 | 75.3 | 74.2 |
| | | NESTER-CL | **70.3** | 60.0 | 72.2 | 70.7 | **75.6** | **67.7** | 77.5 | 76.3 | **76.5** | 69.5 | 78.5 | 77.3 |
| | | NESTER-L3 | 58.7 | 55.7 | 71.9 | 67.8 | 66.2 | 65.9 | 78.4 | 74.9 | 66.7 | 67.7 | 79.5 | 75.9 |
| **Para-metric (%)** | ST | Hityper | 8.4 | 52.7 | 70.2 | 56.5 | 8.4 | 52.7 | 70.2 | 56.5 | 8.4 | 52.7 | 70.2 | 56.5 |
| | DL | TypeBERT | 29.8 | 41.4 | 54.0 | 48.1 | 36.0 | 55.9 | 58.0 | 53.5 | 37.7 | 60.8 | 61.2 | 56.5 |
| | | TypeWriter | 54.4 | 54.1 | - | - | 63.4 | 63.5 | - | - | 68.8 | 69.3 | - | - |
| | | UniXcoder | 61.9 | 61.8 | 44.3 | 49.3 | 72.3 | 76.0 | 51.2 | 57.6 | 75.0 | **80.1** | 53.8 | 60.4 |
| | | Type4Py | 68.0 | 59.0 | **86.2** | 80.2 | 74.1 | 64.1 | 88.3 | 83.3 | 75.9 | 66.3 | 88.8 | 84.3 |
| | LMs | InCoder-1.3B | 22.9 | 22.8 | 18.7 | 19.9 | 23.3 | 23.1 | 19.1 | 20.3 | 23.4 | 23.3 | 19.2 | 20.4 |
| | | InCoder-6.7B | 28.8 | 51.6 | 25.0 | 28.1 | 29.3 | 52.1 | 25.3 | 28.5 | 29.4 | 52.5 | 25.3 | 28.6 |
| | | CodeT5-base | 54.8 | 66.7 | 27.7 | 36.6 | 62.9 | 74.2 | 34.4 | 43.6 | 65.6 | 76.4 | 37.1 | 46.3 |
| | | CodeT5-large | 61.4 | **69.4** | 55.7 | 58.0 | 66.8 | 74.3 | 61.2 | 63.5 | 68.9 | 76.2 | 63.7 | 65.9 |
| | | Naive-CL | 33.9 | 26.0 | 33.3 | 32.9 | 51.5 | 45.5 | 50.1 | 50.1 | 56.2 | 48.1 | 54.7 | 54.6 |
| | | Naive-L3 | 32.6 | 44.2 | 43.9 | 41.3 | 46.4 | 53.2 | 59.9 | 56.3 | 47.2 | 54.5 | 61.0 | 57.4 |
| | | TypeGen-CL | 68.4 | 62.0 | 80.1 | 79.0 | 78.0 | 66.4 | 85.5 | 82.4 | **81.9** | 70.7 | 86.2 | 84.1 |
| | | TypeGen-L3 | 64.9 | 59.5 | 68.9 | 67.2 | 78.5 | 74.5 | 83.1 | 81.6 | 80.3 | 77.5 | 85.1 | 83.6 |
| | | NESTER-CL | **75.7** | 64.1 | 85.3 | **81.5** | **80.0** | 73.8 | 90.5 | **87.2** | 81.7 | 75.8 | 91.5 | **88.2** |
| | | NESTER-L3 | 65.5 | 63.8 | 84.6 | 79.0 | 74.9 | **76.4** | **91.0** | 86.6 | 76.3 | 78.3 | 91.9 | 87.8 |

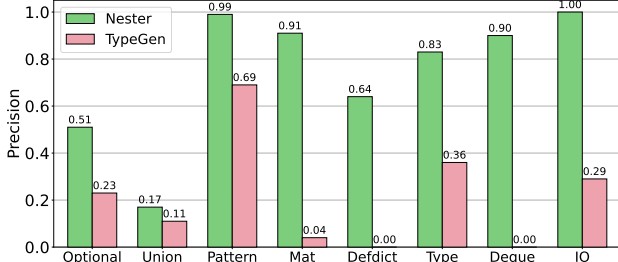

*Figure 5.* Comparison between NESTER and TypeGen.

and 10.9%). This demonstrates NESTER's ability to synthesize control flow. In rarer categories like "typing.Pattern" and "typing.Type", NESTER also excels with scores of 99.4% and 90.0%, compared to TypeGen's 92.0% and 36.4%. These results highlight NESTER's program synthesis capabilities and its ability to extract type information from control flow.

### 4.4. Performance of High-Level Program Generation

To evaluate the accuracy of high-level programs generated by LMs, and to assess the potential of larger LMs for this task, we employ five widely-used models: Code Llama 7B, Llama 3 8B, GPT-3.5, ChatGPT-4, and GPT-4 on 100 Many-Types4Py samples. Three master's students specializing in Computer Science, each with five years of Python programming experience, are invited to evaluate the generated programs. Drawing on prior research in visual question answering (Hu et al., 2024), we developed evaluation criteria for each generated program. Human annotators scored the model responses based on: (1) Correctness: Is the program execution result accurate? (2) Explainability: Does the LM clarify the type inference process? (3) Factuality: Are all steps factually correct compared to the original code snippet? (4) Consistency: Are the results consistent across multiple executions?

Table 2 shows the human evaluation results of five LMs across four metrics. GPT-4 and Llama 3 8B achieve the

*Table 2.* Evaluation of high-level program generation accuracy across different LMs.

| Model | Acc. (%) | Exp. (%) | Fact. (%) | Cons. (%) |
|---|---|---|---|---|
| Code Llama 7B | 80 | 85 | 81 | 73 |
| LLama 3 8B | 96 | 88 | 96 | 100 |
| GPT-4o mini | 95 | 87 | 97 | 95 |
| GPT-4o | 87 | 95 | 91 | 92 |
| GPT-4 | 96 | 87 | 97 | 96 |

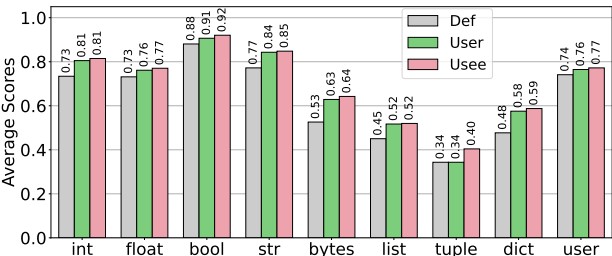

*Figure 6.* Iterative reasoning average accuracy of NESTER method across various data types.

highest correctness, indicating that high-level programs generated by NESTER can be effectively produced by models as small as 8 billion parameters. In contrast, Code Llama 7B achieves only 80% correctness, likely due to Llama 3's superior handling of tasks that combine natural language and code, while Code Llama focuses more on code generation. GPT-4-4o mini scores 95% in explainability but only 87% in overall correctness, suggesting strong generative ability but occasional inaccuracies. This is supported by factuality ratings, where GPT-4 and GPT-4o score highest at 97%, while GPT-4o drops to 91%. Notably, Llama 3 8B achieves 100% consistency, likely due to its lower temperature setting.

### 4.5. Effectiveness of Symbolic Parsing

To evaluate NESTER's symbolic parsing, we focus on the types inferred based on regular expressions. The results, shown in Table 3, highlight NESTER's coverage across local, argument, and return variables in various type categories: "int", "float", "bool", "str", "bytes", "list", "tuple", "dict", and "set". For local variables, types like "int" (54.7%) and "str" (57.4%) show good coverage, while others like "float" (25.3%), "bool" (29.7%), and "list" (33.0%) are also effectively inferred. More complex types, such as "set" (4.3%), are less covered but still show meaningful inference, confirming the validity of symbolic reasoning. For return types, coverage is generally lower (around 3.4%), with function arguments showing minimal coverage (0.1%). This suggests that symbolic reasoning is less effective for return types and arguments due to the higher frequency of local definitions.

### 4.6. Effectiveness of Iterative Inference

To assess the efficacy of NESTER's iterative inference, we conduct experiments on NESTER for each reasoning step.

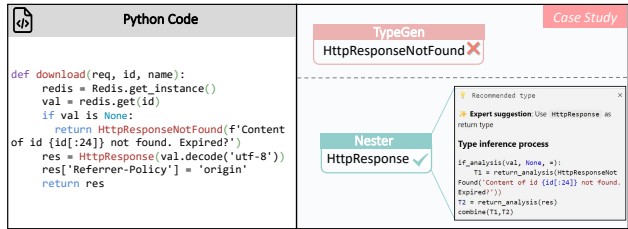

*Figure 7.* Case study of a real-world type inference example using NESTER.

We detail the results of this three-step analysis across nine categories, including "int", "float", "bool", "str", "bytes", "list", "tuple", "dict", and "user_defined", in Figure 6. From this figure, we can see an improvement in accuracy across all categories. Notably, in the step of inference **Def**, NESTER infers over 50% of the types, indicating that initial type inference is relatively easy due to the simplicity of the types. Furthermore, there is a significant performance improvement from **Def** to **User** compared to the transition from **User** to **Usee**. This suggests that the step-by-step inference is particularly effective in the early stages, but as more static knowledge is incorporated, the types become increasingly difficult to infer. We also observe that certain categories, such as "tuple" and "user", do not show substantial improvements from **Def** to **User**. This can be attributed to the specific categories that are most likely defined or utilized during the **Def** stage.

### 4.7. Case Study

To compare our methods with TypeGen, we present one type inference example from real-world scenarios in GitHub repositories, as shown in Figure 7. In this case, which focuses on the Python function *download*, the return type needs to be inferred. TypeGen incorrectly reasons the return type as *HttpResponseNotFound*. In contrast, NESTER first generates a high-level program, incorporating control flow in *if_analysis* and type information for T1 and T2 in *assignment_analysis*. It merges this information based on control flow, and during program interpretation, LMs evaluate if the control flow can proceed. Due to insufficient information, LMs default to allowing the flow to proceed. The assignment of *HttpResponseNotFound* to T1 and *HttpResponse* to T2 is done using LMs, and static analysis reveals that T1 is a subclass of T2, concluding with the final type *HttpResponse*.

## 5. Related Work

**Type Inference**  Type inference reduces the programming burden in dynamic languages by minimizing explicit type annotations. While static-rule-based solutions have been widely used, they often have limited coverage due to the complexity of dynamic languages (Chen & Erwig, 2016;

*Table 3.* Coverage analysis of symbolic rules across different type categories.

| Type Category | int | float | bool | str | bytes | list | tuple | dict | set | total |
|---|---|---|---|---|---|---|---|---|---|---|
| Var. | 54.7 | 25.3 | 29.7 | 57.4 | 5.5 | 33.0 | 2.1 | 0.6 | 4.3 | 32.2 |
| Arg. | 0.1 | 0 | 0 | 0.3 | 0 | 0.09 | 0 | 0.1 | 0 | 0.1 |
| Ret. | 11.1 | 3.9 | 2.2 | 3.5 | 2.3 | 2.2 | 3.1 | 0 | 0 | 3.4 |

Furr et al., 2009; Pavlinovic et al., 2021; Ke et al., 2024). In contrast, machine learning's adaptability has led to the adoption of deep-learning techniques for type inference. These methods are typically divided into two categories: the supervised approach (Allamanis et al., 2020; Pradel et al., 2020; Mir et al., 2021), which treats type inference as a classification task, and the cloze-style approach (Wei et al., 2023), which frames it as a fill-in-the-blank task. The rise of language models like GPT-3, using a chain-of-thought approach (Peng et al., 2023), has further enhanced reasoning and interpretability in type inference tasks.

**LMs for Reasoning**  LMs for reasoning enhance human understanding of problem-solving and excel in complex tasks like deducing formulas, solving logical puzzles, and analyzing data patterns (Wang et al., 2023a; Mizrahi et al., 2023; Biswas & Talukdar, 2024). To unlock their full zero-shot potential, techniques such as prompt tuning, in-context learning, chain-of-thought reasoning, and Retrieval-Augmented Generation (RAG) have been developed. While effective in simpler tasks, LMs struggle with complex reasoning, often producing 'hallucinations' or inaccurate outputs (Tonmoy et al., 2024; Ji et al., 2023; Yao et al., 2023). In RAG, for example, LMs may be influenced by retrieved examples, leading to flawed reasoning. Compared to human cognition, LMs still require significant improvement in their reasoning abilities. This limitation becomes particularly evident when the model size is reduced from hundreds of billions to tens of billions of parameters.

**Neural-Symbolic Reasoning**  Neural-symbolic learning combines deep learning's representational power with the logical rigor of symbolic reasoning to address complex cognitive tasks. In 2018, Manhaeve et al. (2018) introduced DeepProbLog, integrating neural networks with probabilistic logic programming to enhance reasoning. VIS-PROG (Gupta & Kembhavi, 2023) leverages LMs for visual reasoning tasks without task-specific training, highlighting their potential. For more complex visual tasks, Embed2Sym utilizes symbolic optimization via embedding space clustering to enhance reasoning efficiency (Aspis et al., 2022). Furthermore, Manhaeve et al. (2018) extended neural-symbolic learning to natural language processing using advanced techniques such as inductive logic programming. Despite these advancements, challenges persist in integrating deep learning with symbolic systems, particularly in terms of scalability and robustness.

## 6. Discussion

Our work adopts a neuro-symbolic approach to type inference by decomposing the task into simpler steps. We further enhance inference by integrating both dataflow and symbolic information into LMs, allowing for more accurate and interpretable type predictions. Despite these clear advantages, the practical integration of symbolic components introduces certain limitations.

A key limitation of our approach is its restricted applicability to larger-scale LLMs. Although NESTER, integrated with the symbolic rules on the Code Llama 7B, successfully inferred types that Code Llama 7B alone could not handle, larger models like Code Llama 70B are able to infer some of these types without the constraints of symbolic rules, as models with larger parameters generally possess better comprehension and generalization capabilities. In future work, we plan to extend NESTER to LMs of larger scale. We also intend to adapt our approach to more locally deployable models, such as Microsoft's Phi series, to explore a balance between inference accuracy and deployment efficiency.

Another limitation is the difficulty in extending our approach to other programming languages. Although we utilized NESTER to tackle the issue of dynamic type inference within Python, integrating it into Java presented significant challenges due to Java's strict syntax; for instance, Java requires explicit type declarations at variable initialization and prohibits null values without explicit nullable declarations (Lanzinger et al., 2021). Consequently, we will undertake a comprehensive redesign when adapting the type inference system designed for Python to Java, ensuring that high-level programs are restructured to meet Java's syntactic and type compatibility requirements.

## 7. Conclusion

We have presented NESTER, a new neuro-symbolic method for type inference. NESTER leverages LMs to acquire high-level programs through in-context learning. Subsequently, it employs a program interpreter to execute the program. By harnessing the dataflow information within the program, NESTER facilitates human-like reasoning. Moreover, each interface is designed to be extensible and offers good interpretability. Experimental results demonstrate the effectiveness of NESTER and highlight the usefulness of its major modules.

## Acknowledgements

This work is supported by the National Key Research and Development Program of China under Grant No. 2022YFB4501400, the National Natural Science Foundation of China under Grant No. 62372199, and the Major Program (JD) of Hubei Province (Grant No. 2023BAA024).

## Impact Statement

The proposed framework, NESTER, adopts a neuro-symbolic approach to enhance the type inference capabilities of LMs. By integrating dataflow-guided reasoning, our method tackles key challenges in type inference, resulting in more accurate and reliable predictions. The high-level program we introduce not only improves type inference but also expands to other areas of code intelligence, including code summarization, code generation, and potentially automated debugging. This approach is part of the community's ongoing efforts in combining traditional software engineering techniques with the powerful capabilities of modern LMs. Additionally, our framework paves the way for developing more robust, interpretable, and scalable LMs for software development, ultimately improving both the efficiency and safety of code-related tasks. The broader impact of this work extends to a wide range of applications, from improving developer productivity to advancing research in LM-assisted programming and automated software engineering.

Although NESTER, when integrated with symbolic rules on Code Llama 7B, successfully inferred types that Code Llama 7B alone could not, larger models such as Code Llama 70B are capable of inferring some of these types independently, further demonstrating the potential of combining symbolic reasoning with large-scale LMs.

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

# A. Preliminaries

This section presents several preliminaries essential to our work, including LMs, type inference, and neuro-symbolic reasoning.

## A.1. Language Models

In the past few years, LLMs such as the GPT series (Koubaa, 2023) and Llama series (Das et al., 2025b) have made significant progress in text and code generation (Naveed et al., 2023; Das et al., 2025a). LMs aim to predict the probability distribution of the next token based on a given sequence of preceding tokens. Let $s$ denote a sentence or code sample tokenized into a sequence $\{w_1, w_2, \ldots, w_{|s|}\}$. The LM estimates the joint probability $p(s)$ as the product of conditional probabilities:

$$p(s) = \prod_{i=1}^{|s|} p(w_i \mid w_1, w_2, \ldots, w_{i-1}).$$

Training is performed on a large corpus $\mathcal{D} = \{s_1, s_2, \ldots, s_N\}$ of size $N$, using the negative log-likelihood loss function:

$$\mathcal{L}_{\text{LM}}(\theta) = -\sum_{s \in \mathcal{D}} \sum_{i=1}^{|s|} \log p(w_i \mid w_1, w_2, \ldots, w_{i-1}; \theta),$$

where $\theta$ represents the model parameters. Once trained, the language model can generate subsequent tokens during inference by iteratively selecting tokens with the highest predicted probabilities from the vocabulary.

The emergence of LLMs is transforming the learning paradigm from the traditional "pre-train and fine-tune" to "pre-train, prompt, and predict" (Liu et al., 2023). Instead of extensively fine-tuning LMs for each downstream task, tasks are now reframed using textual prompts to align with the models' original training objectives. Various prompting techniques have been developed to fully leverage LMs' capabilities.

To maximize the potential of LMs, researchers have developed various prompting techniques that reframe tasks to align with the models' original training objectives. **Zero-shot prompting** (Kojima et al., 2022) leverages well-designed prompts to guide LMs in handling new tasks without additional training data, such as generating class hierarchies in game design or solving math problems. **In-context learning** enhances task comprehension by providing a few input-output examples, enabling the model to infer solutions based on similar cases. For more complex reasoning, **Chain-of-Thought (CoT) prompting** (Wei et al., 2022; Zhang et al., 2024) breaks down problems into step-by-step reasoning processes, improving accuracy in tasks like arithmetic. Finally, **Least-to-Most (LTM) prompting** further refines CoT by decomposing difficult problems into progressively simpler subproblems, ensuring robust and scalable reasoning. These techniques collectively optimize LM performance across diverse applications.

## A.2. Type Inference

The type inference task studied in this work is to predict the type of an identifier based on its relevant context within a code snippet. In this paper, we focus on Python, a dynamically-typed language, which is characterized by untyped programs but can utilize an existing deterministic type system. Given a code snippet $c$, let $\{x_v \mid v \in 1 \ldots V\}$ represent the set of $V$ distinct identifiers within $c$ that require type assignment. Let $p(\tau_i \mid ctx)$ denote the probability distribution(may not probability distribution) over types for an identifier $x_v$, determined by the context $ctx$ of the code. Our objective is to develop a type inference model that can input an entirely or partially unannotated code snippet $c$ and output a probability distribution of types for each missing annotation. Typically, the prediction space $\mathcal{Y}(c)$ consists of two components: the set of all user-defined types (classes/interfaces) declared within $c$ and a fixed set of commonly-used library types. Let $\mathcal{T}$ represent the typing environment of the code snippet $c$, and $\mathcal{T} \vdash c$ indicate that the program $c$ is well-typed according to the deterministic type system, with types for identifiers specified by $\mathcal{T}$.

**LM-based Type Inference** Recently, as the LMs have achieved substantial progress in content generation, including text generation (OpenAI, 2023), code generation (Lu et al., 2023), and video synthesis (Wu et al., 2023), several studies have leveraged the capabilities of LLMs for type inference (Siddiqui & Kellogg, 2024; Peng et al., 2023). The key of LM-based type inference approaches lies in framing the task as a generative one by designing a natural language prompt. As shown in Figure 8, the prompt describes the given code and identifies the relevant identifiers for type prediction. Such a prompt

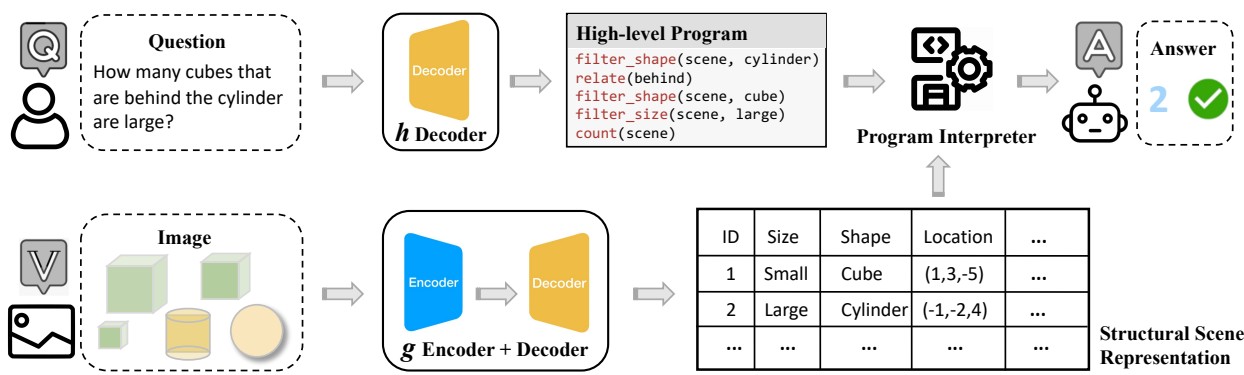

*Figure 9.* Neuro-symbolic visual question answering.

can effectively guide LMs to perform type inference. TypeGen (Peng et al., 2023) represents the current state-of-the-art in LM-based type inference, employing a specialized prompt designed around ICL. The TypeGen prompt is structured as [Code, ICL EXAMPLES, TYPEHINTS], where ICL EXAMPLES refers to demonstration examples used for ICL, and TYPEHINTS denotes prediction hints extracted through prior static analysis techniques. Additionally, TypeGen incorporates CoT reasoning to enhance the accuracy of type inference. The CoT prompt in TypeGen is structured as follows: "*First, the variable* `error_message` *is assigned from a string. Therefore, the type of the variable* `error_message` *is* '**str**'. "

### A.3. Neuro-Symbolic Reasoning

Neuro-symbolic reasoning is an innovative approach that enhances the interpretability of tasks by breaking them down into manageable subtasks (Gupta & Kembhavi, 2023; Cunnington et al., 2022; Nye et al., 2021; Zhang et al., 2021). Specifically, it divides the solution function, denoted as $f$, into two distinct phases:

$$f = h \circ g. \qquad (1)$$

Here, $g$ represents a perception network that processes raw data through human-like constraints to establish a set of conditions for solving the problem. $h$, on the other hand, functions as a reasoning model that analyzes the raw data to delineate a logical inference pathway, leading to the solution. Neuro-symbolic reasoning aims to optimize the learning of these two functions to identify the most appropriate $f$.

The neuro-symbolic reasoning methodology has demonstrated significant promise across a range of applications, such as visual reasoning (e.g., visual question answering) (Gupta & Kembhavi, 2023), mathematical problem-solving (Wang et al., 2023b), and code search (Arakelyan et al., 2018).

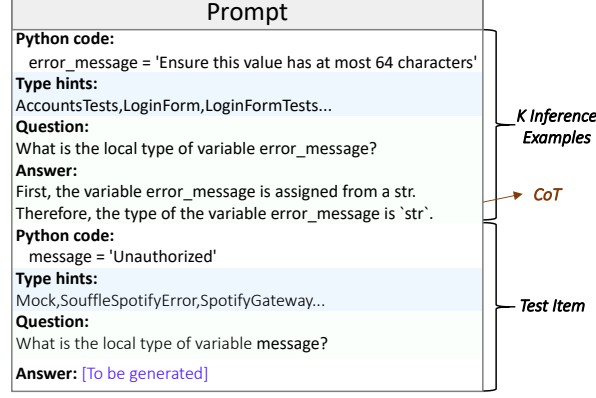

*Figure 8.* An examplar prompt of LMs for type inference (Peng et al., 2023).

***An Example of Visual Reasoning***    Figure 9 shows an example of neuro-symbolic visual question answering, the goal of which is to derive an answer to a specified question by analysing the provided image. Consider the query "*How many cubes that are behind the cylinder are large?*". For the perception function $g$, the image is processed into a structure analogous to a dataframe, which serves as static information for addressing the question. As for the reasoning function $h$, it employs a context-aware model to parse the question text into a high-level program. This model identifies the key terms of the question specifically, *cube*, *cylinder*, and *large*—which are utilized as parameters for the high-level program. The program then generates a sequence of steps to address the question using the provided APIs, such as *filter_shape* and *count*.

Finally, this high-level program is executed by a program interpreter, utilizing the database created earlier by the perception function $g$ before producing the final result answer *2*. It is noteworthy that if errors occur in the high-level program, the program interpreter will randomly sample an answer from all possible outputs of the final module. In this case, the sampling

will be limited to a numerical range that is contextually appropriate for the query, such as the number of cubes depicted in the image.

## B. Details of High-Level Program Generation

### B.1. High-Level Program Definition

**High-Level Program Definition** We define the structure of a high-level program formally, starting by modeling the original program as a flow graph $G = (V, E)$, where $V$ represents nodes and $E$ represents control flows. Intra-procedurally, a high-level program is represented as a flow graph $G' = (V', E')$, where each node $V$ in $G$ corresponds to a smaller flow graph $G''(V'', E'')$ within $G'$. Each node in $V'$ is defined by a function $F : G \to G''$ that selects a subgraph from $G$ and represents it as a node in $G'$. This ensures that $\bigcup V' = G$ and that for any two nodes $v_i, v_j \in V'$, $v_i \cap v_j = \emptyset$. We define control flows within $V'$ as *Sub-Control Flow* $(\mathcal{F}_1^{sub}, \mathcal{F}_2^{sub}, \dots, \mathcal{F}_n^{sub})$, and edges $e \in E'$ as *Main-Control Flow* $(\mathcal{F}^{main})$, reflecting their hierarchical relationship.

### B.2. Prompts for High-Level Program Generation

**Local Type Prompt** The **local type prompt** focuses on transforming Python code that involves local assignments or expressions. The goal is to use the transformation API `Assignment_Analysis()` for assignments and `If_Analysis()` for conditional statements to generate a high-level program description for each line of code.

---

**Example:**

```
Code:
urlpatterns = [path('about_project', views.index, name='video')]
High-level Program:
urlpatterns = Assignment_analysis([path('about_project', views.index, name='video')])

Code: buf = self.getvalue()
High-level Program:
buf = Assignment_analysis(self.getvalue())

Code:
If val is None:
  app_name = 'about'
High-level Program:
If_analysis(val, None, is)
  app_name = Assignment_analysis('about')

Code:
If val is None:
  a = 123
else:
  a = 'abc'
High-level Program:
If_analysis(val, None, is)
   a = Assignment_analysis(123)
If_analysis(val, None, is not)
  a = Assignment_analysis('abc')

Code:
metrics_to_return = {}
  if not self.error_analysis and name.startswith('_'):
    if is_empty_metric(metric):
      if isinstance(metric, CategoricalAccuracy):
High-level Program:
metrics_to_return = Assignment_analysis({})
  If_analysis(not self.error_analysis and name.startswith('_'))
    If_analysis(is_empty_metric(metric))
      If_analysis(isinstance(metric, CategoricalAccuracy))
```

---

**Arg Type Prompt** The **arg type prompt** is designed for analyzing functions and arguments. It uses the Function_Analysis() API for functions and the Argument_Analysis() API for specific arguments or values passed into functions. This prompt is intended to break down function definitions and their corresponding arguments. It is useful for situations where the focus is on understanding function signatures and the arguments they receive.

This kind of analysis is particularly valuable in real-world development, especially when dealing with complex codebases or performing code reviews. By thoroughly examining argument types, developers can significantly reduce runtime errors and improve code robustness. Additionally, structured analysis like this helps clarify a function's intended behavior, preventing potential logical flaws. When combined with proper documentation, the benefits multiply—clear docstrings and strict type hints are, after all, the foundation of high-quality code.

---

**Example:**

```
Code:
def boot(self, container):
  provider = container.get(settings.Props.DI_PROVIDER)
High-level Program:
Function_Analysis(boot(self, container))
  Argument_Analysis(provider = container.get(settings.Props.DI_PROVIDER))

Code:
def __init__(self, value):
  self.value = value
High-level Program:
Function_Analysis(__init__(self, value))
  Argument_Analysis(self.value = value)

Code:
def _test_convenience_model_restorer(restorer, convenience_method, placeholder_model,
trained_model, ckpt_id, capsys):
  _check_log(restorer, ckpt_id, capsys)
High-level Program:
Function_Analysis(_test_convenience_model_restorer(restorer,
convenience_method, placeholder_model, trained_model, ckpt_id, capsys))
  Argument_Analysis(_check_log(restorer, ckpt_id, capsys))

Code:
def get_mods_manifest(manifest_url):
  if n == 1:
    return json.loads(get_requests_object(manifest_url).text)
High-level Program:
Function_Analysis(get_mods_manifest(manifest_url))
  If_Analysis(n==1)
    Argument_Analysis(return json.loads(get_requests_object(manifest_url).text))

Code:
def save_modlines(manifest_url, mods_details, mods_path):
  if mod_ids.difference(set(mods_details)):
    modlines[modline] = [mods_details[mod_id]['directory_name'] for mod_id in mod_ids]
High-level Program:
Function_Analysis(save_modlines(manifest_url, mods_details, mods_path))
  If_Analysis(mod_ids.difference(set(mods_details)))
    Argument_Analysis(modlines[modline] = [mods_details[mod_id]['directory_name'] for
    mod_id in mod_ids])
```

---

**Return Type Prompt** The **return type prompt** is used for analyzing functions with return statements. The prompt uses the Return_Analysis() API to describe the function's return value, and Combine() is used for combining multiple return values if the function contains more than one conditional return. This type of transformation focuses on the function's return logic and how the return values are constructed based on conditions.

Understanding return behavior is crucial for debugging and ensuring predictable function outputs—after all, inconsistent return types can lead to nasty surprises down the line. By systematically analyzing return paths, developers can catch edge

cases early, enforce type safety, and maintain cleaner control flow. Whether a function returns a single value, multiple conditional results $(r_1, r_2, ..., r_n)$, or even raises exceptions, the `Return_Analysis()` API helps document and verify expected behavior.

In practice, this means fewer "Why is this function returning `None`?" moments and more confidence in code reliability. The relationship between input parameters and output values becomes more transparent when return types are properly analyzed.

---

**Example:**

```
Code:
return TestApp(app())
High-level Program:
Return_Type1 = Return_Analysis(TestApp(app()))
                    Return_Type = Return_Type1

Code:
return json.loads(get_requests_object(manifest_url).text)
High-level Program:
Return_Type1 = Return_Analysis(json.loads(get_requests_object(manifest_url).text))
Return_Type = Return_Type1

Code:
if n == 1:
  return mods_details
High-level Program:
If_Analysis(n == 1)
  Return_Type1 = Return_Analysis(mods_details)
Return_Type = Return_Type1

Code:
if code != 0:
  return False
if counter == 3:
  return True
High-level Program:
If_Analysis(code != 0)
  Return_Type1 = Return_Analysis(False)
If_Analysis(counter == 3)
  Return_Type2 = Return_Analysis(True)
Return_Type = Combine(Return_Type1, Return_Type2)

Code:
if payload.get('type') == 'auth':
  return json_success({'full_name': user_profile.full_name,
  'email': user_profile.email, 'id': user_profile.id})
if topic is None:
  if topic is None:
    if content is None:
      return json_success()
High-level Program:
If_Analysis(payload.get('type') == 'auth')
  Return_Type1 = Return_Analysis(json_success({'full_name': user_profile.full_name,
  'email': user_profile.email, 'id': user_profile.id}))
If_Analysis(topic is None)
  If_Analysis(topic is None)
    If_Analysis(content is None)
      Return_Type2 = Return_Analysis(json_success())
Return_Type = Combine(Return_Type1, Return_Type2)
```

---

A line of the high-level program, or a program step, consists of the name of an API, the API's input argument names and the API's output variable name. To aid LM generation, we use descriptive API names (e.g., `if_analysis`, `return_analysis`), argument names (e.g., `lr`, `1.0`, `>`), and variable names (e.g., `T1`, `T2`) to allow LM to understand the input and output of each API. The LMs need to identify the locations of these statements (e.g., conditional and

return), fill in the corresponding identifiers from each line as arguments for the respective APIs and create a new variable name as output.

Through this method, we gradually guide from the simplest condition to more complex structures, increasing complexity step-by-step, which is in line with the "least-to-most prompting" instructional strategy. This approach helps the model (or learner) to gradually build up complex logical expressions, enhancing the accuracy and reliability of understanding and execution.

### B.3. Details of Supported Modules in NESTER

NESTER currently supports four modules, including the **If_Statement_Module**, **Assignment_Module**, **Argument_Module**, and **Return_Module**. These modules are designed to implement fundamental constructs and operations of high-level programming languages, facilitating the processing and execution of code within the interpreter. Each module has specific responsibilities and execution workflows, detailed as follows:

▷ **If_Statement_Module**: This module handles conditional statements. It first determines the truth value of the condition expression through the `condition_judgment` method. If the condition is true, the `execute` method will carry out the code within the conditional block; otherwise, it will skip it.

▷ **Assignment_Module**: Responsible for handling variable assignment operations. This module parses variables and expressions in assignment statements using the `symbolic_parse` method, then calculates the value of the expression and updates the corresponding variable in the state dictionary through the `execute` method.

▷ **Argument_Module**: This module manages the passing of arguments in function calls. During the `execute` process, it parses the incoming arguments and calculates each parameter's value based on the current state dictionary, subsequently passing these values to the respective function or procedure.

▷ **Return_Module**: Handles the return operations from functions or methods. By parsing return statements using the `symbolic_parse` method, this module calculates the return value in the `execute` method, updates the state dictionary, and returns the result to the caller.

## C. Details of Program Interpreter

### C.1. Details of Symbolic Parsing

In the high-level program interpreter, we denote the parameters (always a statement) received by the interpreter as $S$. To determine the type of $S$, we first use symbolic parsing to classify the input $x_v$, requiring type annotation. Figure 11 shows the denotational semantics of regular expressions used for symbolic parsing in this paper. These include simple types like int, float, bool, str, and bytes, as well as generic types such as tuple, list, and dict. Finally, if $S$ has not been generated by an LM, we parse it to identify $x_v$ for iterative type inference.

Specifically, for statement $S$, we first use regular expressions to classify the type of input $x_v$. For instance, the statement *return "Odd"* is processed by our regular expression and recognized as type *str*. If our designed regular expression fails to recognize the type, we proceed to parse it, such as in the statement *return a + b*, which we identify as two identifiers $a$ and $b$.

### C.2. Details of CGDG

**Coarse-Grained Dataflow Graph (CGDG)**    To build a bridge from a non-structured code representation to a structured intermediate code graph representation, a Coarse-Grained Dataflow Graph (CGDG) is introduced in NESTER framework. Compared to traditional Concrete Syntax Tree (CST) and Abstract Syntax Tree(AST), the CGDG has at most three hops for each identifier, resulting in a more concise graph structure. The CGDG is constructed from the CST of the target program. For an identifier $x_v$, we represent these three hops using three types of nodes: **Def**$(x_v)$, **User**$(x_v)$, and **Usee**$(x_v)$, while the edges are directed to capture the relationships between them.

For an identifier $x_v$, **Def**$(x_v)$ represents the definition of $x_v$ within the code snippet $c$. This includes all top-level assignment statements where $x_v$ appears on the left-hand side. Let $\mathcal{U}_c$ represent the set of all code constructs, *i.e.*, the individual nodes in the CST that correspond to elements such as variable declarations and operations. **User**$(x_v)$ includes all constructs $u$ in which the identifier $x_v$ is used for computations or condition checks. We define **User**$(x_v) = \{u \in \mathcal{U}_c :$

```python
class If_Statement_Module():
  def __init__(self):
    # load a trained model; move to GPU

  def condition_judgment(self, step: str):
    # parse the list of input expression/variable
      names/operator and output a bool value of the condition

  def execute(self, step: str, state: Dict):
    step += query_table(query)
    inputs, input_var_names, output_var_name = self.parse(step)

    # get values of input variables from state
    for var_name in input_var_names:
      inputs.append(state[var_name])

    # perform computation using the loaded LLMs
    query, type = some_computation(inputs)

    # update state
    state[output_var_name] = output
    return output
```

```python
class Assignment_Module():
  def __init__(self):
    # load a trained model; move to GPU

  def symbolic_parse(self, step: str):
    # parse step and return list of input values/variable
      names/operator and output the type or variable name

  def execute(self, step: str, state: Dict):
    inputs, input_var_names, output_var_name = self.parse(step)

    # get values of input variables from state
    for var_name in input_var_names:
      inputs.append(state[var_name])

    # perform computation using the loaded LLMs
    type = some_computation(inputs)

    # update state
    state[output_var_name] = output
    return output
```

```python
class Argument_Module():
  def __init__(self):
    # load a trained model; move to GPU

  def execute(self, step: str, state: Dict):
    step += query_table(query)
    inputs, input_var_names, output_var_name = self.parse(step)

    # get values of input variables from state
    for var_name in input_var_names:
      inputs.append(state[var_name])

    # perform computation using the loaded LLMs
    query, type = some_computation(inputs)

    # update state
    state[output_var_name] = output
    return output
```

```python
class Return_Module():
  def __init__(self):
    # load a trained model; move to GPU

  def symbolic_parse(self, step: str):
    # parse step and return list of input values/variable
      names/operator and output the type or variable name

  def execute(self, step: str, state: Dict):
    step += query_table(query)
    inputs, input_var_names, output_var_name = self.parse(step)

    # get values of input variables from state
    for var_name in input_var_names:
      inputs.append(state[var_name])

    # perform computation using the loaded LLMs
    query, type = some_computation(inputs)

    # update state
    state[output_var_name] = output
    return output
```

*Figure 10.* Modules currently supported in NESTER.

$$\llbracket \text{int} \rrbracket = \{w \mid w = s \cdot d,\ s \in \{+, -\},\ d \in \llbracket \text{digits} \rrbracket\}$$

$$\llbracket \text{float} \rrbracket = \{w \mid w = s \cdot d_1.d_2,\ s \in \{+, -\},\ d_1, d_2 \in \llbracket \text{digits} \rrbracket\}$$

$$\llbracket \text{string} \rrbracket = \{w \mid w = s_1 \ldots s_n,\ s_i \in (\sigma \cup \epsilon),\ n \geq 0\}$$

$$\llbracket \text{bool} \rrbracket = \{w \mid w = \text{``}True\text{''} \lor w = \text{``}False\text{''}\}$$

$$\llbracket \text{bytes} \rrbracket = \{w \mid w = b(s_1 \ldots s_n),\ s_i \in (\sigma \cup \epsilon),\ n \geq 0\}$$

$$\llbracket \text{list} \rrbracket = \{w \mid w = [[s_1, s_2]],\ s_1, s_2 \in \llbracket \text{string} \rrbracket\}$$

$$\llbracket \text{dict} \rrbracket = \{w \mid w = \{[s_1, s_2]\},\ s_1, s_2 \in \llbracket \text{string} \rrbracket\}$$

$$\llbracket \text{set} \rrbracket = \{w \mid w = \{s_1, s_2, \ldots, s_n\},\ s_i \in \llbracket \text{string} \rrbracket,\ n \geq 0\}$$

$$\llbracket \text{tuple} \rrbracket = \{w \mid w = ([s_1, s_2]),\ s_1, s_2 \in \llbracket \text{string} \rrbracket\}$$

$$\llbracket \text{digits} \rrbracket = \{w \mid w = d_1 d_2 \ldots d_n,\ d_i \in \llbracket \text{digit} \rrbracket,\ n \geq 1\}$$

$$\llbracket \text{digit} \rrbracket = \{w \mid w \in \{0, 1, 2, 3, 4, 5, 6, 7, 8, 9\}\}$$

$$\llbracket \epsilon \rrbracket = \{w \mid w \text{ is a valid escape sequence}\}$$

$$\llbracket \sigma \rrbracket = \{w \mid w \text{ is any valid character}\}$$

*Figure 11.* Denotational semantics of regular expressions applied in NESTER.

$x_v$ is actively utilized in $u$}, where active utilization indicates that the identifier $x_v$ is involved in computational operations or evaluations. **Usee**$(x_v)$ contains not just the direct users of $x_v$, but also anything that is used in the user context **Usee**$(x_v) = \bigcup_{u \in \textbf{User}(x_v)} \{v \mid v \text{ is involved in } u\text{'s computation}\}$, $v$ denotes a variable actively engaged in $u$'s operations.

---

**Algorithm 1** Coarse-grained Dataflow Graph Generation from CST

---

**Input:** The CST of the function, $func\_cst$
**Output:** Coarse-grained dataflow graph of the given function, $cg$
**for** $n \in func\_cst$ **do**
   **if** $n.type = Token \wedge n.tokenType = Identifier$ **then**
      **if** hasCSTDef($n$) **then**
         $cg$.addNode(Def($n$), $n.line\_number$)
      **end**
   **end**
**end**
**for** $n \in cg$ **do**
   $cst\_node \leftarrow$ getCSTNode($n$) **if** hasCSTUser($cst\_node$) **then**
      $user\_node \leftarrow$ getCSTUser($cst\_node$) **if** **$Def$**($user\_node$) $\in cg$ **then**
         $cg$.addEdge(Def($n$), Def($user\_node$), 0) $cst\_user\_node \leftarrow$ getCSTNode($user\_node$) **if** hasCSTUser($cst\_user\_node$)
        **then**
           $usee\_node \leftarrow$ getCSTUser($cst\_user\_node$) **if** $usee\_node \in cg$ **then**
              $cg$.addEdge(**Def**($n$), **Def**($usee\_node$), 1)
           **end**
           **else**
              $cg$.addNode(**Usee**($n$), $usee\_node.line\_number$) $cg$.addEdge(**Def**($n$), **Usee**($n$), 1)
           **end**
        **end**
      **end**
      **else**
         $cg$.addNode(User($n$), $user\_node.line\_number$) $cg$.addEdge(Def($n$), User($n$), 0)
      **end**
   **end**
**end**

---

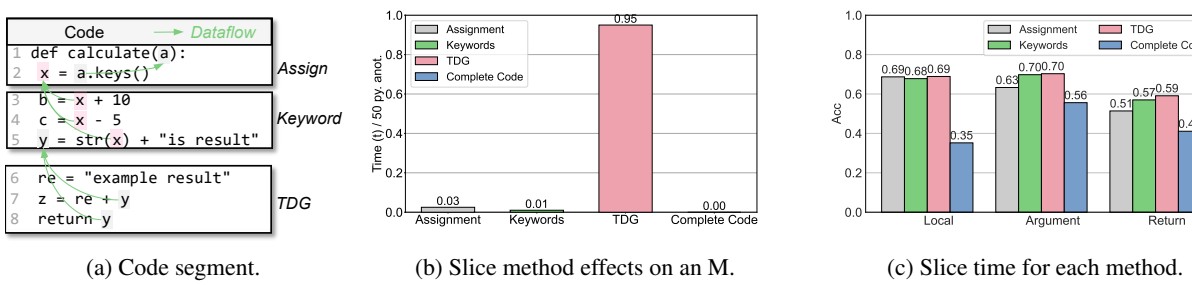

(a) Code segment.      (b) Slice method effects on an M.      (c) Slice time for each method.

*Figure 12.* Impact of code slice on type inference performance.

**CGDG Construction** Given the input code's CST, NESTER constructs a CGDG for each function following the logic in Alg. 1. It first identifies all identifier definitions $Def(x_v)$ and builds CGDG nodes by traversing the CST. The RedBaron library is used to visit each CST node. When an identifier node is found (line 2), NESTER creates a corresponding CGDG node and preserves the line number (line 4). The process repeats recursively until all identifier nodes are visited, reaching the CST's leaf nodes (line 6).

After node instantiation, NESTER traverses the CGDG. As each node is encountered, it identifies the corresponding CST node and checks for identifiers in use (line 9). If a node uses another, NESTER checks for an existing matching node in the CGDG. If found, an edge is added (line 12). Otherwise, a new **User**($n$) node is created, and an edge is added from **Def**($n$) to **User**($n$) (line 24). Similarly, **Usee**($n$) nodes are created if necessary, and a Usee edge is added from **Def**(n) to **Usee**($n$) (line 21).

# D. Extended Analysis: Rethinking Type Dependency Graph (TDG) for Neural Type Inference

To the best of our knowledge, TypeGen (Peng et al., 2022; 2023) is the first to explore LMs for type inference using a Type Dependency Graph (TDG) to model variable type dependencies and static analysis for code slicing to improve efficiency. We argue that the TDG may be too detailed, impacting inference efficiency. To test this, we design an experiment to assess the TDG's role and the granularity of code slices. As shown in Figure 12a, code can be sliced at different granularities,

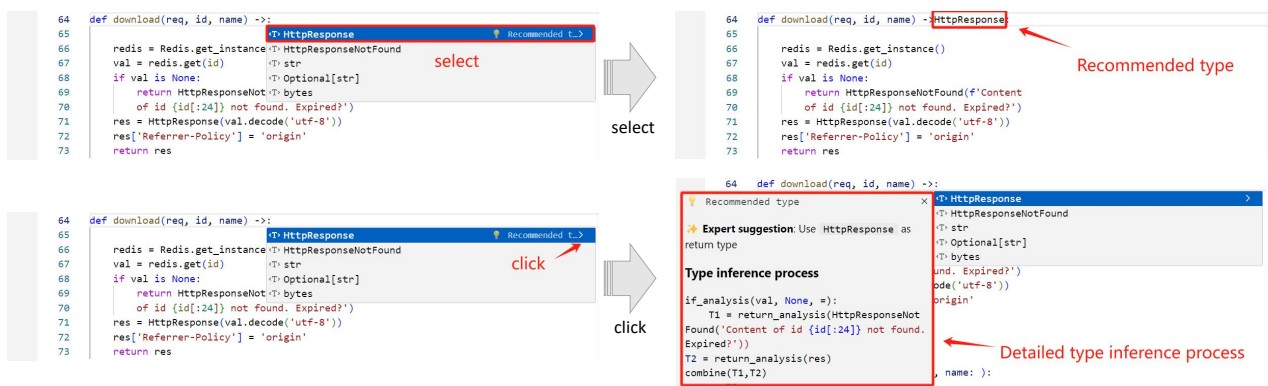

*Figure 13.* Example from case study 1 solved by the NESTER plugin. You can directly select the type recommended by NESTER to complete the annotation. Alternatively, you can click the '>' icon to view the detailed type inference process.

such as by assignments, keywords, or TDG. These slices are then input into an LM (Code Llama 7B) to predict variable, argument, and return types.

Figure 12b shows that slicing by Assignment, Keywords, and TDG all improve accuracy compared to using the full code. The TDG method achieves the highest accuracy, but for local types, the Assignment method is nearly as accurate ( 69%). For argument types, the Keywords method reaches 70%, close to the TDG's performance. Figure 12c compares slicing times for 50 Python annotations. The TDG method takes an average of 0.95 seconds, while the Assignment and Keywords methods take just 0.02 and 0.05 seconds, respectively—20 times faster. These results show that while the TDG performs well, it sacrifices time, even underperforming simple keyword searches. We suggest that LMs don't need such detailed static analysis; coarser slices can still provide good results with greater efficiency. Moreover, a new method is needed to better integrate LMs with static analysis to avoid reducing the potential of combining both.

## E. Implementation

To show the interpretability of our method, we developed a plugin on Visual Studio Code (VSCode). Any developer can download the plugin to their local system. This plugin is crafted to assist developers in obtaining type inference results and gaining a deeper understanding of the type inference process. We use simple regular expressions to generate high-level programs, which can not handle complex code structures, but fortunately can handle most simple code, thereby facilitating a seamless interaction.

Figure 13 depicts a screenshot of the VSCode plugin. The plugin's interface is designed to be intuitive and interactive, featuring an arrow-based type annotation mechanism. When developers append the '− >' symbol after a function definition for type annotation, the plugin promptly activates a feedback interface. This interface displays five potential types associated with the annotated identifier. For instance, download is predominantly identified as an HttpResponse type. Additionally, a 'click' option is available beneath the recommended type, allowing developers to view the inference process if desired. Upon clicking, a detailed high-level program response is displayed, which elucidates the concrete reasoning steps of the type inference process.

