# OpenReview forum: "Dataflow-Guided Neuro-Symbolic Language Models for Type Inference"
_ICML.cc/2025/Conference — ICML 2025 poster_

### Official Review · Reviewer_f94C · 2025-03-03

**Overall Recommendation:** 3

**Summary:**

The authors present a framework for enhancing language models’ ability to accurately infer types from code. This is achieved by decomposing the input into a high-level program composed of evaluations and type analyses using LMs. This high level program can then be deterministically executed to perform type inference.

**Claims And Evidence:**

The authors primary claims are that their proposed framework improves type inference by applying smaller models (suitable for local deployment) in a structured manner. The claim is supported by their evaluation, which reports accuracy of various approaches on various tasks. The results of this evaluation indicate that their approach achieves greater accuracy in general, compared against other symbolic and neural methods.

**Essential References Not Discussed:**

To my knowledge, all relevant references are included.

**Experimental Designs Or Analyses:**

The experimental design and analysis is valid. Mild concerns are raised in a later section of this review, but by and large there are enough results presented to justify the claims made in the paper. I am referring specifically to Table 1 (the primary results table) and the subsequence analysis in section 4.

**Methods And Evaluation Criteria:**

Both the evaluation criteria and methodology make sense for the application; the authors use program generation and dataflow analysis to reconstruct the input as a series of modules to capture dependencies. This ‘program’ is passed to an interpreter for execution

**Other Comments Or Suggestions:**

n/a

**Other Strengths And Weaknesses:**

Strengths:
+ Broad range of LMs included in the evaluation.
+ Detailed methodology makes the approach clear.
+ Appendix addresses information omitted in the main body.
+ Experiments and code are made available for reproducibility.

Weaknesses:
+ Only one purely symbolic method is included in the evaluation.
+ Only one target language is used in evaluation.

**Questions For Authors:**

1. How much more expensive is NESTER inference in terms of time and energy over the naive CL and L3 models?

2. Is NESTER extensible to compiled languages, or are interpreted languages the only candidate for type inference with NESTER?

3. Why not evaluate on manytypes4typescript as well? It seems that NESTER should at least be extensible to JS/TS.

**Relation To Broader Scientific Literature:**

This paper contributes to the trend of producing neuro-symbolic frameworks which apply a deep learning model(s), usually large language models, in a structured system that executes symbolic operations or algorithms on programs or data synthesized by the model. This paper supports an argument for this design pattern, which is that certain abilities that are strong in truly large models but weaker in smaller models can be achieved by the smaller models if this design pattern is applied correctly.

**Theoretical Claims:**

There do not seem to be any rigorous theoretical claims or proofs in the main body of the paper.

---

> ### Author Rebuttal · Authors · 2025-04-01
>
> We thank the reviewer for the positive feedback.
> * * *
> **Q1:** How much more expensive is Nester inference in terms of time and energy over the naive CL and L3 models?
>
> **A1:** We conduct additional experiments to evaluate Nester's computational cost in terms of inference time and energy consumption using an NVIDIA RTX 4090 GPU. The results are as follows:
>
> | Method                                      | Time (s) | Energy (J)  | Accuracy (%) |
> |---------------------------------------------|---------:|------------:|-------------:|
> | Naïve-CL                                    |    4910  |     219,317 |         31.3 |
> | Nester-CL                                   |   18,396 |   1,365,714 |         68.7 |
> | Nester-CL (w/o multi-step reasoning)        |    4391  |     228,023 |         65.7 |
> | Naïve-L3                                    |     436  |      20,818 |         42.4 |
> | Nester-L3                                   |    2906  |     233,378 |         63.6 |
> | Nester-L3 (w/o multi-step reasoning)        |     465  |      22,248 |         59.6 |
>
> Nester incurs higher computational costs, primarily due to multiple invocations of large LMs during inference. However, when multi-step reasoning is removed, the additional overhead is negligible, making the inference time and energy consumption comparable to those of the naive models. In this case, Nester will only locate the relevant code lines (e.g., identifier definitions) from the source code according to the high-level program. It then leverages LLMs along with predefined rules in a single step to infer types without further iteration, still yielding practical results for type inference. This provides an optional solution with fast type inference time.
> * * *
> **Q2:** Is Nester extensible to compiled languages, or are interpreted languages the only candidate for type inference with Nester?
>
> **A2:** This is a good point. Nester can be adapted to lower-level compiled languages like bytecode and assembly instructions. This will require adjusting the high-level program prompts for LMs and developing language-specific rules for inference, taking into account the existing type systems of the specific languages. We will add a discussion.
> * * *
> **Q3:** Why not evaluate on ManyTypes4TypeScript as well? It seems that Nester should at least be extensible to JS/TS.
>
> **A3:** Our current implementation does not support certain JS/TS features due to the limitations of the underlying tools used by Nester. These include JavaScript's dynamic scoping (e.g., eval), prototype-based inheritance, event-driven asynchrony, and TypeScript's hybrid semantics, such as gradual typing and any escape hatches—none of which are present in Python. We will clarify this and discuss how to extend Nester to other languages beyond Python.
> * * *

---

### Official Review · Reviewer_fVw9 · 2025-03-09

**Overall Recommendation:** 4

**Summary:**

The paper presents Nester, a novel neuro-symbolic technique for type inference. Nester decomposes the type inference process into sub-tasks that are aligned with the data and control flows of the input code, encapsulating these into a modular high-level program. This program executes multi-step actions, such as evaluating expressions and analyzing conditional branches, thereby integrating static typing with language models (LMs) to deduce potential types. Implemented for Python, Nester is benchmarked against various models, including symbolic, deep learning-based, and LM-based systems, using the ManyTypes4Py dataset. Although Nester does not surpass Type4Py in exact match accuracy, it excels in both exact match and match to parametric metrics for inferring argument and return types. Additionally, Nester is available as a VSCode plugin, providing users with visibility into the high-level program generated by the LM and insight into the LM's reasoning process.

**Claims And Evidence:**

The paper presents an interesting claim regarding whether LMs perform reasoning in type inference. The experiments conducted in Section 2.3 provide convincing evidence to support this claim.

**Essential References Not Discussed:**

The related works are sufficiently discussed in this paper.

**Experimental Designs Or Analyses:**

The experimental design is reasonable. The paper first presents the overall performance, then separately discusses the neural and symbolic rule-based components, and finally provides a case study. Additionally, it examines the effectiveness of high-level programs.

**Methods And Evaluation Criteria:**

The proposed method, Nester, makes sense as it builds upon existing work in both rule-based and LM-based approaches. The evaluation criteria are convincing since the dataset, ManyTypes4Py, is commonly used in other studies, and the metrics—exact match and match to parameters—are standard for type inference task [1], [2].

[1] Peng, Yun, et al. "Generative type inference for python." 2023 38th IEEE/ACM International Conference on Automated Software Engineering (ASE). IEEE, 2023.

[2] Guo, Yimeng, et al. "Generating Python Type Annotations from Type Inference: How Far Are We?." ACM Transactions on Software Engineering and Methodology 33.5 (2024): 1-38.

**Other Comments Or Suggestions:**

None

**Other Strengths And Weaknesses:**

Pros:

1. This paper introduces a novel methodology that seamlessly combines static analysis (symbolic) with neural perceptual representations (neural) via LMs to enhance the accuracy and effectiveness of type inference tasks. To the best of my knowledge, this work is the first to apply a neuro-symbolic approach in software engineering, specifically selecting type inference as a suitable application scenario.
2. This paper reveals the limitations of traditional rule-based methods and recent LMs-based approaches by providing a motivating example. Motivated by this, the authors introduce a neuro-symbolic system that addresses the shortcomings that neither rule-based nor LMs-based methods can fully resolve. Additionally, a practical and easy-to-understand example is used throughout the paper to clearly illustrate its motivation and approach.
3. This paper presents a high-level program interpreter that initially conducts data flow analysis on the target program to construct a data flow graph, which is reasonable and effective.
4. To validate the effectiveness of the proposed Nester, the authors have conducted experiments and compared Nester against eight other systems, including three types of models: rule-based models, deep-learning models, and LM-based models.

Cons:
1. Nester seems to mainly use LLMs for type inference. How does it differ from other LLM-based methods, such as TypeGen?
2. I would assume that LM-based high-level program generation might sometimes produce errors. How does the proposed Nester handle situations when the generated program is not entirely correct?
3. Although the authors provide an example of a high-level program generation in the paper, a specific definition is lacking. Could the author provide a detailed formalization of high-level programs?
4. One limitation of the approach is that Nester is only implemented for very simple code structures, neglecting complex code structures. Are there challenges in implementing it for more complex code structures?

**Questions For Authors:**

1. I would like to understand your contributions in the area of type inference. How does your work differ from existing LLM-based approaches, such as TypeGen?
2. Could the author please provide a detailed formalization of high-level programs?
3. In the context of this dataflow graph, what specifically has been made coarse? How does this compare to the original dataflow graph?
4. It seems that Nester employs a symbolic execution method for type inference. How does Nester compare to traditional symbolic execution approaches?
5. Since rule-based methods are designed, there must be concerns about coverage. What is the coverage of your rules?

**Relation To Broader Scientific Literature:**

This paper presents a novel neuro-symbolic method for addressing the type inference task, specifically focusing on tackling control flow.

**Theoretical Claims:**

The paper does not provide specific theoretical proofs

---

> ### Author Rebuttal · Authors · 2025-04-01
>
> **Q1:** The key differences between Nester and TypeGen.
>
> **A1:** Please see  **Q2** of **Reviewer Q6Mu** for a discussion.
> * * *
> **Q2:** Could the author please provide a detailed formalization of high-level programs?
>
> **A2:** This is provided in Appendix D.1. We will better highlight.
> * * *
> **Q3:** Comparison between the introduced coarse-grained dataflow graph and the original one.
>
> **A3:** Thanks. This information is provided in Appendix E.2 in the context of CGDG (Coarse-Grained Dataflow Graph). Specifically, for an identifier $x_v$, we represent its three hops using three types of nodes: **Def**($x_v$), **User**($x_v$), and **Usee**($x_v$). In essence, our approach differentiates nodes based on their hop distance from $x_v$, categorizing them into three distinct types. Compared to traditional Concrete Syntax Tree (CST) and Abstract Syntax Tree (AST), CGDG constraints each identifier to at most three hops, resulting in a more concise and structured graph representation. We will explain this approach in more detail.
> * * *
> **Q4:** How does Nester compare to traditional symbolic execution approaches?
>
> **A4:** Nester is a static analysis framework for type inference and does not rely on symbolic execution—no symbolic execution traces are generated. Unlike traditional symbolic execution, which explores possible input spaces to produce traces, Nester operates purely on static source code information. We will clarify.
> * * *
> **Q5:** What is the coverage of the rules in Nester?
>
> **A5:** The coverage of rules is reported in **Table 3**. Our rules support primitive types (int, float, bool, str, bytes) and container types (list, tuple, dict, set),  covering the vast majority of type inference scenarios encountered in real-world code.
> * * *

---

> > ### Comment · Reviewer_fVw9 · 2025-04-01
> >
> > I thank the authors for resolving my concerns.

---

### Official Review · Reviewer_q6Mu · 2025-03-13

**Overall Recommendation:** 1

**Summary:**

The paper introduces a neuro-symbolic approach named NESTER, which integrates language models (LMs) with program analysis for type inference in dynamically typed languages like Python. With the help of LMs, NESTER translates target code into a high-level program composed of predefined API-driven analysis units and thus decomposes type inference into modular sub-tasks guided by
dataflow analysis. This program is executed via a neuro-symbolic interpreter that combines static analysis (e.g., coarse-grained dataflow graphs) and LM-based reasoning to infer types. It uses a type recognizer to initially identify types from explicit type declarations in the code. When the type recognizer fails, two language models are involved: a condition-evaluation language model to assess conditional statements in the high-level program and a type-inference language model to infer types based on context and the dataflow graph. Evaluated on the ManyTypes4Py dataset using metrics like Exact Match and Match to Parametric, NESTER outperforms the selected baselines by 3-4% in Top-1 accuracy.

## update after rebuttal

The authors did not provide enough evidence to show that the proposed method can perform well for user defined types or rare types, which is known as a common chanllenge in the type inference task. Moreover, the authors also did not provide how they construct the datasets from ManyTypes4Py. Based on this, I have concerns about whether the datasets are cherry-picked from ManyTypes4Py.

**Claims And Evidence:**

The authors claim their approach to be the first neuro-symbolic framework for type inference using LMs. The existing work TypeGen also follows combines static analysis and LLM. What are the key differences?

**Essential References Not Discussed:**

The references are adequate.

**Experimental Designs Or Analyses:**

The experimental design is generally acceptable, with some issues which will be discussed in the weakness part.

**Methods And Evaluation Criteria:**

The proposed methods are evaluation criteria are reasonable.

**Other Comments Or Suggestions:**

- While the case study highlights NESTER’s ability to correct TypeGen’s errors (e.g., inferring HttpResponse instead of HttpResponseNotFound), it does not analyze failure modes.
- The authors may consider to submit the paper to a software engineering venue, as the key contributions seems to be the high-level program generation and program interpreter parts.

**Other Strengths And Weaknesses:**

Strengths:
- NESTER effectively bridges neural and symbolic paradigms by leveraging LMs as neural sub-task parsers while retaining control-flow semantics through symbolic program decomposition.
- The paper provides empirical analyses of practical challenges in LM-based type inference. For instance, it investigates the pitfalls of in-context learning (ICL), demonstrating that LM accuracy drops by 8.5% when exposed to misleading demonstrations.
- The use of LMs to generate high-level programs avoids reliance on language-specific parsers, suggesting scalability to other dynamically typed languages (e.g., JavaScript).

Weaknesses:
- In the motivation example, it's more like that adding an unreachable statement to fool the CodeLlama model (from float—none to bool—float—none).
- While NESTER reports strong results on common and Union-like types (e.g., 53.2% accuracy on `typing.Optional`), its evaluation lacks dedicated metrics for complex parametric types (e.g., `Dict[str, List[int]]`) and user-defined types. This contrasts with some previous work [1][2][3], which explicitly benchmarks rare/complex/user-defined types. This omission raises concerns about generalizability as real-world codebases heavily utilize such types, especially given ManyTypes4Py’s known dataset distribution bias on rare type, which was reported in [2].
- The experimental improvement is relatively marginal.

References
[1] Allamanis, M. et al. "Typilus: Neural Type Hints." PLDI 2020.
[2] Wei, J. et al. "TypeT5: Seq2seq Type Inference using Static Analysis." ICLR 2023.
[3] Wang, X. et al. "TIGER: A Generating-Then-Ranking Framework for Practical Python Type Inference." ICSE 2024.

**Questions For Authors:**

How does the proposed method perform for complex parametric types and user-defined types?

**Relation To Broader Scientific Literature:**

This paper is an increment compared to previous work, as the proposed method differs from existing work.

**Theoretical Claims:**

NA

---

> ### Author Rebuttal · Authors · 2025-04-01
>
> **Q1:** How does the proposed method perform for complex parametric types and user-defined types?
>
> **A1:** In response to your concern, we will provide Nester's performance on **complex parametric types** (depth = 0, e.g., list; depth = 1, e.g., list[str]; depth ≥ 2, e.g., Dict[str, List[int]]) and **user-defined types**, as follows.
>
> | Method           | Depth=0 | Depth=1 | Depth>=2 | User-defined |
> |------------------|---------|---------|----------|--------------|
> | Exact Match      |         |         |          |              |
> | TypeGen-CL       | 40.8    | 39.3    | 7.0      | 74.7         |
> | Nester-CL        | 58.3    | 30.7    | 2.1      | 81.3         |
> | TypeGen-L3       | 35.7    | 31.3    | 2.1      | 69.3         |
> | Nester-L3        | 55.8    | 26.8    | 2.1      | 78.7         |
> | Match to Para.   |         |         |          |              |
> | TypeGen-CL       | 68.3    | 69.3    | 35.5     | 75.0         |
> | Nester-CL        | 79.9    | 75.2    | 58.7     | 81.2         |
> | TypeGen-L3       | 51.5    | 60.5    | 31.3     | 69.5         |
> | Nester-L3        | 66.3    | 73.8    | 61.7     | 78.6         |
>
> From this table, we can observe that Nester outperforms Typegen for user-defined types in terms of exact match and match to parametric metrics. Furthermore, Nester performs better with complex parametric types in terms of match to parametric metrics. Although Nester does not achieve higher exact match accuracy for depth ≥ 2 and depth=1, this can be attributed to Typegen's use of specially designed chain-of-thought operations for recursive type structures, which Nester has not yet incorporated. We believe that integrating such techniques into Nester could further enhance its performance, and we will explain how this can be done later.
> * * *
> **Q2:** The key differences between Nester and TypeGen (raised in the **Claims and Evidence** part).
>
> **A2:** Nester differs from TypeGen in its approach to type inference. Nester integrates static data and control flow analysis with the capability of LLMs.  In contrast, TypeGen relies entirely on LLM-generated reasoning, treating code analysis as solely textual processing progress—a less reliable approach and more prone to hallucination. Our design also allows Nester to provide an intuitive, high-level program view, offering greater interpretability during type inference than TypeGen.
> * * *

---

> > ### Comment · Reviewer_q6Mu · 2025-04-03
> >
> > I still have concerns about the provided experimental results. From the tables, the results of user-defined types are even better than those of built-in types (e.g., list, set, dict as indicated by the results of the depth=0 column). This is quite strange. Can the authors provide explanations for this?

---

> > > ### Author Response · Authors · 2025-04-03
> > >
> > > Thank you for getting back to us. Our approach achieves better performance on user-defined types by leveraging type hints derived from import analysis, which are included in the LLM prompt (see also Appendix A.2). These hints help the LLM make more accurate predictions for user-defined types.
> > >
> > > To ensure a fair comparison, we report performance separately for user-defined and built-in types. We will clarify this in the final version.
> > >
> > > Please let us know if you have any further questions.

---

### Official Review · Reviewer_ugzA · 2025-03-13

**Overall Recommendation:** 4

**Summary:**

This paper proposes Nester, a neurosymbolic tool for performing type inference with LLMs. It uses LLMs to generate high-level versions of the code in question, and determines the return type of a function by analyzing its data and control flow. It outperforms existing SOTA type inference tools for simple as well as complex types.

**Claims And Evidence:**

I believe the claims are clear and the paper shows the evidence to support them.

**Essential References Not Discussed:**

Not that I am aware of.

**Experimental Designs Or Analyses:**

The experiments seem sound. As for the results, I would have liked a better explanation for why Type4Py exceeds performance in certain cases. The paper says that Type4Py is stronger in handling short code segments, but is it just based on lines of code in the function? There are also cases where CodeT5-Large exceeds performamce, so an explanation for that would be nice too.

Apart from that, the evaluation section tables use a lot of abbreviations which were not specified earlier (e.g. Arg., Ret., Var. in table 1), so please add definitions for those (unless I missed them somewhere in the paper).

Also, what are the main challenges preventing you from extending support to other datasets? Based on the paper, it should be relatively easy to just apply to any function defined in Python.

**Methods And Evaluation Criteria:**

Yes, the paper presents a good evaluation with relevant baselines for the problem.

**Other Comments Or Suggestions:**

1. Have you thought about evaluating Nester with small language models like Microsoft's Phi family?
2. In Figure 3, for line 6 in the high level program, why is only T2 and T3 being combined? The condition evaluation hasn't been run yet and it doesn't know T1 is unreachable, so should it not assume for now that T1, T2 and T3 should all be combined?

**Other Strengths And Weaknesses:**

In general, the problem being tackled here is both important and difficult to solve, and the proposed technique is novel and elegant. The paper is also well explained and clear on its contributions.

I do believe, despite what is written in threats to validity, that Nester should be evaluated over larger LLMs. What is the fundamental reason why Nester couldn't just be run with a larger LLM? It would be interesting to see if Nester + LLama-70B performs better than just LLama 70B, or if the gains made by the neurosymbolic reasoning diminish with larger models. I know the main motivation was to use smaller LLMs, but section 4.2 suggests an openness to evaluating Nester with larger models.

The paper also does not have a discussion on limitations of Nester wrt the input functions itself. For instance, does Nester only work with individual functions and local context, or can it draw from contexts outside the scope of the function? E.g., if there was a function call to some function custom defined in a repository, will Nester assume the return type of the called function or in turn perform type-inference for that function? Other statistics about functions input to Nester would also give a good idea about its abilities, such as how many lines of code are there in each input?

**Questions For Authors:**

See above.

**Relation To Broader Scientific Literature:**

This paper offers a marked improvement over previous SOTA type-inference methods, and uses techniques relevant to both the type-inference fields as well as the neurosymbolic fields.

**Theoretical Claims:**

Didn't verify.

---

> ### Author Rebuttal · Authors · 2025-04-01
>
> We thank the reviewer for the positive feedback.
> * * *
> **Q1:** What is the fundamental reason why Nester couldn't just be run with a larger LLM?
>
> **A1:** Nester can work with LLMs of various sizes. In this work, we use a modest-size LLM as we target scenarios where users prefer running models locally—on a developer PC or cluster—to avoid sending data to the cloud due to security and privacy concerns. In the final version, we will demonstrate how Nester can be integrated with a larger LLM (70B).
> * * *
> **Q2:** Does Nester only work with individual functions and local context, or can it draw from contexts outside the scope of the function?
>
> **A2:** Our current implementation performs type inference on individual functions and local context. This can be easily extended by - for example, inlining the callee functions. We will provide a discussion.
> * * *
> **Q3:** Other statistics about the functions input to Nester, such as the number of lines of code in each input.
>
> **A3:** Agree. The distribution of lines of code in our dataset is as follows.
>
> | Line Range     | Count  |
> |---------------|-------:|
> | ≤ 5 lines     | 68,860 |
> | 6–10 lines    | 13,108 |
> | 11–15 lines   | 2,340  |
> | 16–20 lines   | 557    |
> | > 20 lines    | 340    |
>
> We will provide this information.
> * * *
> **Q4:** Have you thought about evaluating Nester with small language models like Microsoft's Phi family?
>
> **A4:** Thanks for your comments. Based on your suggestion, we simply evaluate Nester on 100 instances from ManyTypes4Py using top-1 predictions with Phi-3.5-mini-instruct, due to the limited time slot. The results are as follows:
>
> | Method             | Arg.  | Ret. | Var. | All  |
> |--------------------|------|------|------|------|
> | Exact Match       |      |      |      |      |
> | Naive            | 27.3 | 72.7 | 39.4 | 40.4 |
> | Nester           | 22.7 | 72.7 | 51.5 | 47.5 |
> | Match to Para.   |      |      |      |      |
> | Naive            | 31.8 | 72.7 | 43.9 | 44.4 |
> | Nester           | 22.7 | 72.7 | 60.6 | 53.5 |
>
> From this table, we can see that Nester outperforms the naive method in overall accuracy. However, it does not surpass the naive method in argument (Arg.) and return(Ret.) type predictions. This is likely due to the small number of return and argument types in the 100-instance dataset—if a single inference fails, it can result in a significant deviation. We leave further investigation of Nester on Microsoft's Phi family for future work.
> * * *
> **Q5:**  In Figure 3, for line 6 in the high level program, why is only T2 and T3 being combined?
>
> **A5:** In the paper, we simplified the presentation by combining T2 and T3 in Figure 3. In actual execution, however, the interpreter processes T1, T2, and T3 separately and checks whether the returned variable is empty. For this example, the runtime check confirms that T1 is unreachable. We will clarify.
> * * *

---

### Decision · Program_Chairs · 2025-05-01

**Decision:**

Accept (poster)

**Comment:**

This paper presents a neurosymbolic approach to type inference in dynamically typed languages. NESTER translates the code into a high-level program which can then be processed by a neurosymbolic interpreter. The evaluation suggests that results are improved over a large number of baselines.

Overall, the reviewers felt that this work proposes a novel technique that works well and the problem is interesting and relevant to the community. Thus, I recommend that this paper is accepted.